# COPULA CONFORMAL PREDICTION FOR MULTI-STEP TIME SERIES FORECASTING

## ABSTRACT

Accurate uncertainty measurement is a key step to building robust and reliable machine learning systems. Conformal prediction is a distribution-free uncertainty quantification algorithm popular for its ease of implementation, statistical coverage guarantees, and versatility for underlying forecasters. However, existing conformal prediction algorithms for time series are limited to single-step prediction without considering the temporal dependency. In this paper we propose a **Copula** **C**onformal **P**rediction algorithm for multivariate, multi-step **T**ime **S**eries forecasting, **CopulaCPTS**. On several synthetic and real-world multivariate time series datasets, we show that CopulaCPTS produces more calibrated and sharp confidence intervals for multi-step prediction tasks than existing techniques.

## 1 INTRODUCTION

Deep learning models are becoming widely used in high-risk settings such as healthcare, transportation, and finance. In these settings, it is important that a model produces calibrated uncertainty to reflect its own confidence and to assist decision making. Confidence regions are a common approach to quantify forecast uncertainty (Khosravi et al., 2011). A $(1 - \alpha)$-confidence region $\Gamma^{1-\alpha}$ for a random variable $y$ is *valid* if it contains $y$'s true value with high probability: $\mathbb{P}(y \in \Gamma^{1-\alpha}) \geq 1 - \alpha$. Note one can make $\Gamma^{1-\alpha}$ infinitely wide to satisfy validity. For the confidence region to be useful, we want to minimize its area while remaining valid; this is known as the *efficiency* of the region.

Conformal prediction is a powerful method that produces confidence regions with finite-sample guarantees of validity (Vovk et al., 2005; Lei et al., 2018). Furthermore, it makes no assumptions about the forecast model or the underlying data distribution. Its generality, simplicity, and statistical guarantees have made conformal prediction popular for many real world applications including drug discovery (Eklund et al., 2015) and robotics (Luo et al., 2021).

In this paper we present a more calibrated and efficient conformal prediction algorithm for multi-step times series forecasting. This type of problem is ubiquitous - examples include predicting hurricane paths, vehicle trajectories, and financial or epidemic forecasts. To quantify uncertainty, we want a "cone of uncertainty" that covers the entirety of the forecast horizon. Stankevičiūtė et al. (2021) presented an algorithm, CF-RNN, for multi-step time series with the assumption that all time steps are modeled independently. In practice, however, we found that CF-RNN produces confidence regions often too large to be useful, especially when the problem is multivariate or has a long forecast horizon.

We introduce **CopulaCPTS**, a **Copula**-based **C**onformal **P**rediction algorithm for multi-step **T**ime **S**eries forecasting. We improve efficiency by utilizing copulas to model the dependency between forecasted time steps. A copula is a multivariate cumulative distribution function that models the dependence between multiple random variables. It is widely used in economic forecasting (Nelsen, 2007; Patton, 2012) and has been introduced to conformal algorithms by Messoudi et al. (2021) to model correlation between multiple targets in non-temporal settings. Copula processes have been explored in generative models for time series (Salinas et al., 2019; Drouin et al., 2022). We found that copulas are effective in capturing uncertainty relations between time steps. Our contributions are:

- **CopulaCPTS** is a general uncertainty quantification algorithm that can be applied to *any* multivariate multi-step forecaster, with statistical guarantees of validity.
- **CopulaCPTS** produces significantly sharper and more calibrated uncertainty estimates than state-of-the-art baselines on 4 benchmark datasets, both synthetic and real.

- We present a re-calibration technique such that **CopulaCPTS** can produce valid confidence intervals for time series forecasts of varying lengths.

## 2  RELATED WORK

**Uncertainty Quantification for Deep Time-Series Forecasting**   The two major paradigms of Uncertainty Quantification (UQ) methods for deep neural networks are Bayesian and Frequentist. Bayesian approaches estimate a distribution over the model parameters given data, and then marginalize these parameters to form output distributions via Markov Chain Monte Carlo (MCMC) sampling (Welling & Teh, 2011; Neal, 2012; Chen et al., 2014) or variational inference (VI) (Graves, 2011; Kingma et al., 2015; Blundell et al., 2015; Louizos & Welling, 2017). Wang et al. (2019); Wu et al. (2021) propose Bayesian Neural Network (BNN)-based models for UQ of spatiotemporal forecasts. In practice, Bayesian UQ has two major drawbacks: (1) it requires significant modification to the training process of deep models, and (2) BNNs can be computationally expensive and difficult to optimize, especially for larger networks (Lakshminarayanan et al., 2017; Zadrozny & Elkan, 2001). UQ for deep neural network time series forecasts often adopt approximate Bayesian inference such as MC-dropout (Gal & Ghahramani, 2016b; Gal et al., 2017).

Frequentist UQ methods emphasize robustness against variations in the data. These approaches either rely on resampling the data or learning an interval bound to encompass the dataset. For time series forecasting UQ, frequentist approaches include ensemble methods such as bootstrap (Efron & Hastie, 2016; Alaa & van der Schaar, 2020) and jackknife methods (Kim et al., 2020; Alaa & Van Der Schaar, 2020); interval prediction methods include interval regression through proper scoring rules (Kivaranovic et al., 2020; Wu et al., 2021), and quantile regression (Takeuchi et al., 2006), with many recent advances for time series UQ (Tagasovska & Lopez-Paz, 2019; Gasthaus et al., 2019; Park et al., 2022; Kan et al., 2022). Many of the frequentist methods produces asymptotically valid confidence regions and can be categorized as distribution-free UQ techniques as they are (1) agnostic to the underlying model and (2) agnostic to data distribution.

**Conformal Prediction.**   Conformal prediction (CP) is an important member of distribution-free UQ methods; we refer readers to Angelopoulos & Bates (2021) for a comprehensive introduction and survey. CP has become popular because of its simplicity, theoretical soundness, and low computational cost. A key difference between CP and other UQ methods is that under the exchangeability assumption, conformal methods guarantee validity in finite samples (Vovk et al., 2005).

Most relevant to our work is recent progress on expanding conformal prediction to time-series forecasting. According to Stankevičiūtė et al. (2021) there are two settings: when the data is generated from one single time series or from multiple independent time series. For the first setting, ACI (Gibbs & Candes, 2021) and EnbPI (Xu & Xie, 2021) developed CP algorithms that relaxes the exchangeability assumption while maintaining asymptotic validity via online learning (former) and ensembling (later); Zaffran et al. (2022) further improves on the online adaptation, and Sousa et al. (2022) combines EnbPI with conformal quantile regression (Romano et al., 2019) to model heteroscedastic time series. These algorithms for single time series are not designed to model multiple independent time series where extracting common patterns can improve forecast and UQ results. The validity guaranteed by these methods are on *average* over time steps and are often asymptotic, rather than covering the full horizon as in our setting. Stankevičiūtė et al. (2021) shares with us the multi-step forecasting for multiple time series setting, though they focuses on univariate medical time series. We show that their method of applying Bonferroni correction produces inefficient confidence regions, especially when data is multidimensional. Neeven & Smirnov (2018) and Messoudi et al. (2020) are CP algorithms for multi-target regression in the non-temporal setting, creating box-like regions to account for the correlations between the labels. In our work we develop CoupulaCPTS for multivariate multi-step time series forecasting.

## 3  BACKGROUND

### 3.1  INDUCTIVE CONFORMAL PREDICTION (ICP)

Let $\mathcal{D} = (z_1, \ldots, z_n)$ be a dataset of size $n$. We denote $z_i = (x_i, y_i)$ as a sample of an input and output pair that follows the distribution $\mathcal{P}$, where $i$ is the data index. The input space $\mathbf{X}$ and

target space $\mathbf{Y}$ can be two arbitrary measurable spaces, their Cartesian product $\mathbf{Z} = \mathbf{X} \times \mathbf{Y}$ is the sample space. We begin the algorithm by splitting the dataset into a proper training set $\mathcal{D}_{train}$ with $|\mathcal{D}_{train}| = m$ and a calibration set $\mathcal{D}_{cal}$ with $|\mathcal{D}_{cal}| = n - m$. The objective of conformal prediction is produce a *valid* confidence region (Definition 1).

**Definition 1** (Validity). *Given a new data pair $(X, Y) \sim \mathcal{P}$ and a desired coverage rate $1 - \alpha \in (0, 1)$, region $\Gamma^{1-\alpha}(x)$ is valid if*

$$\mathbb{P}_{(X,Y)\sim\mathcal{P}}(Y \in \Gamma^{1-\alpha}(X)) \geq 1 - \alpha$$

An important component of conformal prediction is the *nonconformity score* $A : \mathbf{Z} \times \mathbf{Z}^m \to \mathbb{R}$, a function to capture how well a sample $z$ *conforms* to the proper training set. For example, we may choose nonconformity score $A(z, \mathcal{D}_{train})$ to be the L2-loss on the new sample:

$$A(z, \mathcal{D}_{train}) := \|y - \hat{f}(x)\| \tag{1}$$

where $\hat{f} : \mathbf{X} \longrightarrow \mathbf{Y}$ is a forecasting model trained with the proper training set $\mathcal{D}_{train}$. In the following sections, we refer to the nonconformity score of a sample $z$ given forecasting model $\hat{f}$ as $A(z, \hat{f})$ for simplicity. Let $s_i = A(z_i, \hat{f})$ denote the nonconformity score of a sample $z_i$ and let $s_{cal} = \{A(z_i, \hat{f})\}_{z_i \in \mathcal{D}_{cal}}$ denote the set of nonconformity scores of all samples in $\mathcal{D}_{cal}$. We define the empirical $p$-quantile of the set of nonconformity scores $\mathcal{S}$ as:

$$Q(p, \mathcal{S}) := \inf\{s' : (\frac{1}{|\mathcal{S}|} \sum_{s_i \in \mathcal{S}} \mathbb{1}_{s_i \leq s'}) \geq p\} \tag{2}$$

**Definition 2** (Exchangeability). *In a dataset $\{z_1, z_2, \ldots, z_n\}$ of length $n$, any of its $n!$ permutations are equally probable.*

Under the exchangeablility assumption (definition 2), conformal prediction is theoretically guaranteed to be valid (definition 1). Let $z_{n+1} = (x_{n+1}, y_{n+1}) \in \mathbf{Z}$ be a new sample, and let $\alpha \in [0, 1]$ be a chosen significance level (and $1 - \alpha$ the confidence level). Since $D_{cal} \cup \{z_{n+1}\}$ is exchangeable, the probability of $A(z_{n+1})$'s rank among $\{A(z_i)\}_{z_i \in \mathcal{D}_{cal}}$ is uniform. Therefore,

$$\mathbb{P}(A(z_{n+1}, \hat{f}) \leq Q(1 - \alpha, s_{cal} \cup \{\infty\})) = \frac{\lceil |(\mathcal{D}_{cal}| + 1)(1 - \alpha)\rceil}{|\mathcal{D}_{cal}| + 1} \geq 1 - \alpha \tag{3}$$

The conformal confidence region is constructed as in equation 4. We say a sample is *covered* if the true value is in the confidence region $y_{n+1} \in \Gamma^{1-\alpha}(x_{n+1})$. Equation 3 means the probability of $y_{n+1}$ being covered is greater than $1 - \alpha$, hence the confidence region is valid.

$$\Gamma^{1-\alpha}(x_{n+1}) := \{y : A(z_{n+1}, \hat{f}) \leq Q(1 - \alpha, s_{cal} \cup \{\infty\})\} \tag{4}$$

The procedure introduced above is known as *inductive* or *split* conformal prediction, as it splits the dataset into training and calibration sets to reduce the amount of computation (Vovk et al., 2005; Lei & Wasserman, 2012). We introduce it as it is commonly used for uncertainty quantification for machine learning models (Papadopoulos, 2008). Although our method in this paper, CopulaCPTS, is implemented based on inductive CP, it can be easily adjusted for other CP variants.

### 3.2 COPULA AND ITS PROPERTIES

In this paper we will model the dependency between time step's uncertainty with copulas. Copula is a concept from statistics that describes the dependency structure in a multivariate distribution. We can use Copulas to capture the joint distribution for multiple future time steps.

This section will briefly introduce its notations and concepts.

**Definition 3** (Copula). *Given a random vector $(X_1, \cdots X_k)$, define the marginal cumulative density function (CDF) as $F_i(x) = P(X_i \leq x)$, the copula of $(X_1, \cdots X_k)$ is the joint CDF of $(U_1, \cdots, U_k) = (F_1(X_1), \cdots, F_k(X_k))$, meaning*

$$C(u_1, \cdots, u_k) = P(U_1 \leq u_1, \cdots, U_k \leq u_k)$$

*Alternatively*

$$C(u_1, \cdots, u_k) = P(X_1 \leq F_1^{-1}(u_1), \cdots, X_k \leq F_t^{-1}(u_k))$$

In other words, the Copula function captures the dependency structure between the variable $X$s; we can view an $k$ dimensional copula $C : [0,1]^k \to [0,1]$ as a CDF with uniform marginals. A fundamental result in the theory of copula is Sklar's theorem.

**Theorem 1** (Sklar's theorem). *Given a joint CDF as $F(X_1, \cdots, X_k)$ and the marginals $F_i(x)$, there exists a copula such that*

$$F(x_1, \cdots, x_k) = C(F_1(x_1), \cdots, F_k(x_k))$$

*for all $x_i \in [-\infty, \infty]$ and $i = 1, \cdots, k$.*

Sklar's theorem states that for all multivariate distribution functions, there exists a copula function such that the distribution can be expressed using the copula and multiple univariate marginal distributions. To give an example, when all the $X_k$s are independent, the copula function is known as the product copula: $C(u_1, \cdots, u_k) = \Pi_{i=1}^{k} u_i$.

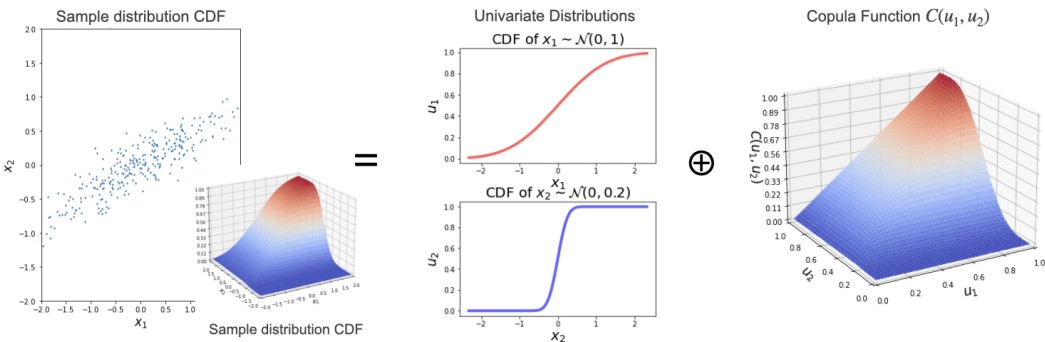

Figure 1: An example copula, where we express a multivarite Gaussian with correlation $\rho = 0.8$ with two univariate distributions and a Copula function $C(u_1, u_2)$.

## 4 COPULA CONFORMAL PREDICTION FOR TIME SERIES (COPULACPTS)

Many real world decision making tasks make use of time series forecasts that are multivariate and can predict multiple steps into the future. In these applications, we want a "cone of uncertainty" that covers the entire course of our forecasts. Existing time series conformal prediction methods either only provide coverage guarantee for individual time step forecasts (Gibbs & Candes, 2021; Xu & Xie, 2021) or produce confidence regions often too inefficient to be useful, especially in multivariate settings (Stankevičiūtė et al., 2021). Hence, we propose a multi-step conformal prediction algorithm for time series, CopulaCPTS, that guarantees validity over multi-step forecasts. We improve efficiency of the confidence regions by modeling dependency of the time steps using a copula function.

We denote the time series dataset as $\mathcal{D} = \{(\mathbf{x}_{1:t}^{(i)}, \mathbf{y}_{t+1:t+k}^{(i)}\}_{i=1}^{l}$, where $\mathbf{x}_{1:t} \in \mathbb{R}^{t \times d_x}$ is $t$ time steps of input which has dimension $d_x$, $\mathbf{y}_{t+1:t+k} \in \mathbb{R}^{t \times d_y}$ is $k$ time steps of output of dimension $d_y$. In the traditional time series forecasting setting we have $d_x = d_y$, but they do not have to be necessarily equal. Given the dataset $\mathcal{D}$, a new test sample $\mathbf{x}_{1:t}^{(n+1)}$, and a confidence level $1 - \alpha$, our algorithm returns $k$ confidence intervals, $[\Gamma_1^{1-\alpha}, \ldots, \Gamma_k^{1-\alpha}]$, one for each time step, such that:

$$\mathbb{P}[\forall h \in \{1, \ldots, k\}, \mathbf{y}_{t+h} \in \Gamma_h^{1-\alpha}] \geq 1 - \alpha \tag{5}$$

for any underlying predictive model. We define equation 5 as the *validity* condition for the multi-step time series setting. We will use superscript $\mathbf{x}^i$ to index data, and subscript $\mathbf{y}_t$ to index the time steps within the multi-horizon $\mathbf{y}$.

There are two characteristics desired in uncertainty quantification methods: *calibration* and *efficiency*. A model is calibrated when the predicted confidence level corresponds to the probability of events falling into the predicted range. It is reflected when equality holds in the validity condition $\mathbb{P}(\mathbf{y} \in \Gamma^{1-\alpha}) = 1 - \alpha$. *Efficiency*, on the other hand, refers to the size of the confidence region. There is a

trade-off between validity and efficiency, as one can always set $\Gamma^{1-\alpha}$ to be infinitely large to satisfy the validity condition. In practice, we want to achieve that the measure of the confidence region (e.g. its area or length) to be as small as possible, given that the validity condition holds. In the following sections we will introduce CopulaCPTS, a conformal prediction algorithm that is both calibrated and efficient for multivariate time series forecasts.

## 4.1 INDUCTIVE CONFORMAL PREDICTION (ICP) FOR MULTIVARIATE FORECASTS

We view the multidimensional target of dimension $d$ as a point in $\mathbb{R}^d$ space, we simply have $\mathbf{Y}_t = \mathbb{R}^d$ for each time step $t$. In this paper we chose nonconformity score to be the L-2 distance $A((x,y),\hat{f}) := \|y - \hat{f}(x)\|$, where $\hat{f}$ is the forecast model trained on the proper training set $\mathcal{D}_{\text{train}}$. The confidence region $\Gamma^{1-\alpha}(\mathcal{D}, x)$ therefore is a $d$-dimensional ball. We chose this metric because we are forecasting trajectories in space. Since the conformal prediction algorithm produces valid confidence regions regardless of the choice of nonconformity score, one can choose other metrics such as Mahalanobis (Johnstone & Cox, 2021) or L-1 (Messoudi et al., 2021) distance based on domain needs, and our algorithm will still hold.

## 4.2 COPULA MULTI-STEP CONFORMAL REGRESSION

We showed in section 3.1 that given a confidence level $1-\alpha$ and a forecast model $\hat{f}$, the ICP algorithm finds a nonconformity score $s_{1-\alpha}$ such that for a new sample $z$ drawn from the data distribution $z \sim \mathcal{P}$:

$$\mathbb{P}(A(z,\hat{f}) \leq s_{1-\alpha}) \geq 1 - \alpha$$

We can use the algorithm to estimate an empirical cumulative distribution function (CDF) for the random variable $A(z,\hat{f})$.

$$\hat{F}(s) := \mathbb{P}(A(z,\hat{f}) \leq s) = \frac{1}{|D_{cal}|} \sum_{z^i \in D_{cal}} \mathbb{1}_{A(z^i,\hat{f}) \leq s} \tag{6}$$

For the multi-step conformal prediction algorithm, we will estimate this empirical CDF for each step of the time series, denoted as

$$\hat{F}_h(s_h) := \mathbb{P}(A(z_h,\hat{f}) \leq s_t) \text{ for } h \in 1, \ldots, k \tag{7}$$

Let $1 - \alpha_h$ denote the probability produced by the each time step's CDF function $F_h(s_h)$, $h \in 1, \ldots, k$. By Sklar's theorem, we have:

$$F(s_1, \ldots, s_k) = C(F_1(s_1), \ldots, F_k(s_k)) = C(1 - \alpha_1, \ldots, 1 - \alpha_k) \tag{8}$$

We want to find the set of confidence levels $1 - \alpha_h$, such that the entire predicted time series trajectory is covered by the intervals with confidence $1 - \alpha$, i.e. $C(1 - \alpha_1, \ldots, 1 - \alpha_k) \geq 1 - \alpha$. The purpose of using copula is to model the dependency between the multiple predicted time steps, so we can better capture the confidence region for the joint probability. This allows our algorithm to produce more efficient confidence regions. We estimate the copula $C$ with the calibration set $\mathcal{D}_{cal}$, and then we can search for values $\alpha_h$ for each time step to obtain the desired multi-step coverage.

We adopt the empirical copula (Ruschendorf, 1976) as our default copula. One may pick other parametric copula functions, such as the Gaussian copula, to introduce inductive bias and improve sample efficiency when calibration data is scarce. The empirical copula is a non-parametric method of estimating marginals directly from observation, and hence does not introduce any bias. For the joint distribution of a time series with $k$ time steps, the copula of a vector of probabilities $\mathbf{u} \in [0,1]^k$ is defined as

$$C_{\text{empirical}}(\mathbf{u}) = \frac{1}{n-m} \sum_{i=m+1}^{n} \mathbb{1}_{\mathbf{u}^i < \mathbf{u}} = \frac{1}{n-m} \sum_{i=m+1}^{n} \prod_{h=1}^{k} \mathbb{1}_{\mathbf{u}_h^i < \mathbf{u}_h} \tag{9}$$

Where $n - m$ is the size of the calibration set. Here, the $\mathbf{u}_i$s are the cumulative probabilities for each data in the calibration set $\mathcal{D}_{cal}$ with size $n - m$.

$$\mathbf{u}^i = (u_1^i, \ldots, u_k^i) = (\hat{F}_1(s_1^i), \ldots, \hat{F}_1(s_k^i)), \ i \in \{m+1, \ldots, n\}$$

---

**Algorithm 1:** Copula Conformal Time Series Prediction

---

**Input:** Dataset $\mathcal{D} = (x_i, y_i)_{i=1,\ldots,n}$, test input $\mathbf{x}_{1:t}^{n+1}$, target significant level $1 - \alpha$.
**Output:** Confidence regions $\Gamma_1^{1-\alpha}, \ldots, \Gamma_k^{1-\alpha}$.

---

1
2 `// Training`
3 Randomly split dataset $\mathcal{D}$ into training and calibration datasets $\mathcal{D} = \mathcal{D}_{train} \cup \mathcal{D}_{cal}$, where $|\mathcal{D}_{train}| = m$ and $|\mathcal{D}_{cal}| = n - m$.
4 Train $k$-step forecasting model $\hat{f}$ with training set $\mathcal{D}_{train}$.
5 `// Calibration`
6 Initialize set of nonconformity scores for each time step $\mathcal{S}_h = \emptyset$ for $h = 1, \ldots, k$.
7 **for** $(\mathbf{x}_{1:t}^i, \mathbf{y}_{t+1:t+k}^i) \in \mathcal{D}_{cal}$ **do**
8     $\hat{\mathbf{y}}_{t+1:t+k}^i \leftarrow \hat{f}(\mathbf{x}_{1:t}^i)$
9     **for** $h = 1$ **to** $k$ **do**
10         $\mathcal{S}_h \leftarrow \mathcal{S}_h \bigcup \|\hat{\mathbf{y}}_{t+h}^i - \mathbf{y}_{t+h}^i\|$
11     **end for**
12 **end for**
13 Construct CDFs $\hat{F}_1 \ldots \hat{F}_k$ as equation 6.
14 Construct copula $C(\hat{F}_1(\cdot), \ldots, \hat{F}_k(\cdot))$ as equation 9.
15 Search for $s_1^*, \ldots, s_k^*$ such that $C(\hat{F}_1(s_1^*), \ldots, \hat{F}_k(s_k^*)) \geq 1 - \alpha$
16 `// Prediction`
17 $\hat{\mathbf{y}}_{t+1:t+k}^{n+1} \leftarrow \hat{f}(\mathbf{x}_{1:t}^{n+1})$
18 **for** $h = 1$ **to** $k$ **do**
19     $\Gamma_h \leftarrow \{\mathbf{y} : \|\mathbf{y} - \hat{\mathbf{y}}_h^{n+1}\| < s_h^*\}$
20 **end for**
21 **return** $\Gamma_1^{1-\alpha}, \ldots, \Gamma_k^{1-\alpha}$

---

Note that to fulfill the validity condition of Equation 5, we only need to find

$$\mathbf{u}^* = (\hat{F}_1(s_1^*), \ldots, \hat{F}_t(s_k^*)) \text{ such that } C_{\text{empirical}}(\mathbf{u}^*) \geq 1 - \alpha \tag{10}$$

We can find $s_1^*, \ldots, s_k^*$ through any search algorithm. As exhaustive search is exponential to the prediction horizon $k$, we implement the search with stochastic gradient descent using PyTorch. We study the effectiveness of this method in Appendix C.6. The confidence region for each time step is constructed as the set of all $y_t \in \mathbf{y}_t$ such that the nonconformity score is less than $s_t^*$. Algorithm 1 summarizes the CoupulaCPTS procedure. We prove that CopulaCPTS produces valid confidence regions for multi-step time series forecasting (Proposition 1) in Appendix A.

**Proposition 1** (Validity of CopulaCPTS). *The confidence regions provided by CopulaCPTS (algorithm 1) are valid. i.e.* $\mathbb{P}[\forall h \in \{1, \ldots, k\}, \mathbf{y}_{t+h} \in \Gamma_h^{1-\alpha}] \geq 1 - \alpha$.

### 4.3 COPULACPTS IN AUTO-REGRESSIVE FORECASTING

Auto-regressive forecasting is a common framework in time series forecasting. So far, we've been looking at forecasts for a predetermined number of time steps $k$. One can use a fixed length model to forecast for longer horizons $k'$ autoregressively - by taking model output as part of the input. In the conformal prediction setting, we want to not only to autoregressively use the point value forecasts, but also propagate the uncertainty measurement.

If we assume the time series to be *stationary*, then the copula remains the same for any sliding window of $k$ steps, i.e. $C(u_1, \ldots, u_k) = C(u_2, \ldots, u_{k+1})$. Given a model $\hat{f}$ to predict $k$ time steps. Hence, after we've found the $u_1^*, \ldots, u_k^*$ such that $C(u_1^*, \ldots, u_k^*) \geq 1 - \alpha$, we simply have to search for $u_{k+1}^*$ such that $C(u_2^*, \ldots, u_k^*, u_{k+1}^*) \geq 1 - \alpha$. The guarantee proven in Proposition 1 still holds for the new estimation. On the other hand, if the time series is *non-stationary*, we can fit copulas $C_1(u_1, \ldots, u_k), \ldots, C_{k'-k}(u_{k'-k}, \ldots, u_{k'})$, one for each autoregressive prediction, given that we have $k'$ steps of data in our calibration set. This way, we transform the autoregressive problem

into $k' - k$ multi-step problems that can be solved by CopulaCPTS. It follows that each of the autoregressive predictions are valid. Appendix B.3 provides an example scenario where re-estimating the copula is necessary and improves validity.

## 5 EXPERIMENTS

In this section, we show that CopulaCPTS produces more calibrated and efficient confidence regions compared to existing methods on two synthetic datasets and two real-world datasets. We also demonstrate that CopulaCPTS's advantage is more evident over longer prediction horizons. We also show its effectiveness in the autoregressive prediction setting.

**Baselines.** We compare our model with three representative works in different paradigms of uncertainty quantification for neural network time series prediction: the Bayesian-motivated Monte Carlo dropout RNN (MC-dropout) by Gal & Ghahramani (2016a), the frequentist blockwise jackknife RNN (BJRNN) by Alaa & Van Der Schaar (2020), and a conformal forecasting RNN (CF-RNN) by Stankevičiūtė et al. (2021). We use the same underlying prediction model for post-hoc uncertainty quantification methods BJRNN, CF-RNN, and CopulaCPTS. The MC-dropout RNN is of the same architecture but is trained separately, as it requires an extra dropout step during training and inference.

**Metrics.** We evaluate calibration and efficiency for each method. For calibration, we report the empirical coverage on the held-out test set. Coverage should be as close to the desired confidence level $1 - \alpha$ as possible. Coverage is calculated as $coverage_{1-\alpha} = \mathbb{E}_{x,y \sim \mathbf{X} \times \mathbf{Y}} P(\mathbf{y} \in \Gamma^{1-\alpha}(\mathbf{x})) \approx \frac{1}{n} \sum_1^n \mathbb{1}(\mathbf{y}_n \in \Gamma_n^{1-\alpha}(\mathbf{x}_n))$. For efficiency, we report the average area (2D) or volume (3D) of the confidence region. The metric should be as small as possible while being valid (coverage maintains above pre-specified confidence level). The area or volume is calculated as $area_{1-\alpha} = \mathbb{E}_{x \sim \mathbf{X}} \|\Gamma^{1-\alpha}(x)\| \approx \frac{1}{n} \sum_1^n \|\Gamma^{1-\alpha}(x_n)\|$.

### 5.1 SYNTHETIC DATASETS

We first test the effectiveness of our models with two synthetic spatiotemporal datasets - interacting particle systems (Kipf et al., 2018), and drone trajectory following (simulated with PythonRobotics (Sakai et al., 2018)). For the particle simulation we predict $\mathbf{y}_{t+1:t+h}$ where $t = 35$, $h = 25$ and $y_t \in \mathbb{R}^2$; for drone simulation $t = 60$, $h = 10$, and $y_t \in \mathbb{R}^3$. To add randomness to the tasks, we added Gaussian noise of $\sigma = .01$ and $.05$ to the dynamics of the particle simulation, and $\sigma = .02$ to drone dynamics. We generate 5000 samples for each dataset, and split the data by 45/45/10 for train, calibration, and test respectively. Please see Appendix C.1 for forecaster model details.

We visualize the calibration and efficiency of the methods in Figure 2 for confidence levels $1 - \alpha = 0.5$ to $0.99$. We can see that Copula-RNN, the red lines, are more calibrated and efficient compared to other baseline methods, especially in tasks with large noise (particle dataset, $\sigma = 0.05$). We can see that for harder tasks (particle $\sigma = 0.05$, and drone trajectory prediction), MC-Dropout is overconfident, whereas BJ-RNN and CF-RNN produce very large (hence inefficient) confidence regions. This behavior of CF-RNN is expected because they apply Bonferroni correction to account for joint prediction for multiple time steps, which is an upper bound of copula functions. The numbers for confidence level $90\%$ are presented in Table 1. A quantitative comparison of confidence regions for drone simulation can be found in Figure 8 in the appendix.

### 5.2 REAL WORLD DATASETS

**COVID-19.** We replicate the experiment setting of Stankevičiūtė et al. (2021) and predict new daily cases of COVID-19 in regions of the UK. The models take 100 days of data as input and forecast 50 days into the future. We used 200 time series for training, 100 for calibration, and 80 for testing.

**Vehicle trajectory prediction.** The Argoverse autonomous vehicle motion forecasting dataset (Chang et al., 2019) is a widely used vehicle trajectory prediction benchmark. The task is to predict 3 second trajectories based on all vehicle motion in the past 2 seconds sampled at 10Hz. Because trajectory prediction is a challenging task, we utilize a state-of-the-art prediction algorithm LaneGCN (Liang et al., 2020) as the underlying model for CF-RNN and Copula-RNN (details in Appendix C.1). Flexibility of underlying forecasting model is an advantage of post-hoc conformal prediction methods.

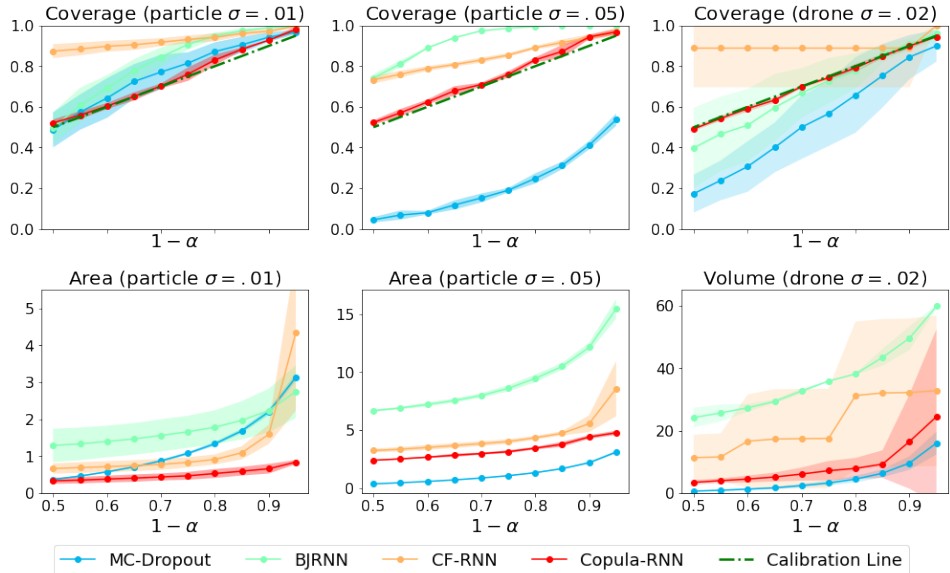

Figure 2: Calibration (upper row) and efficiency (lower row) comparison on different $1 - \alpha$ levels for simulated data sets. For calibration, the goal is to coincide with the green dotted calibration line as closely as possible while staying valid (above the green line). Note that copula methods are more calibrated across different significance levels. For efficiency, we want the metric to be as low as possible. Copula-RNN outperforms the baselines consistently. (MC-dropout for the right two experiments produces invalid regions, so we don't consider its efficiency.)

|  |  | MC-dropout | BJRNN | CF-RNN | Copula-RNN |
|---|---|---|---|---|---|
| Particle Sim | Coverage | 63.9 ± 16.1 | 98.9 ± 0.2 | 97.0 ± 2.3 | **91.5** ± 2.1 |
| ($\sigma = .01$) | Area | 0.81 ± 0.05 | 2.24 ± 0.59 | 3.13 ± 3.24 | **1.06** ± 0.36 |
| Particle Sim | Coverage | 16.1 ± 4.3 | 100.0 ± 0.0 | 94.5 ± 1.5 | **90.3** ± 0.7 |
| ($\sigma = .05$) | Area | 0.79 ± 0.02 | 12.13 ± 0.39 | 5.79 ± 0.51 | **4.50** ± 0.07 |
| Drone Sim | Coverage | 84.5 ± 10.8 | 90.8 ± 2.8 | 91.6 ± 9.2 | **90.0** ± 1.5 |
| ($\sigma = .02$) | Volume | 9.64 ± 2.13 | 49.57 ± 3.77 | 32.18 ± 13.66 | **16.52** ± 7.08 |
| COVID-19 | Coverage | 19.1 ± 5.1 | 79.2 ± 30.8 | 95.4 ± 1.9 | **92.1** ± 1.0 |
| Daily Cases | Area | 34.14 ± 0.84 | 823.3 ± 529.7 | 610.2 ± 96.0 | **429.0** ± 15.1 |
| Argoverse | Coverage | 27.9 ± 3.1 | 92.6 ± 9.2 | 98.8 ± 1.9 | **90.4** ± 0.3 |
| Trajectory | Area | 127.6 ± 20.9 | 880.8 ± 156.2 | 396.9 ± 18.67 | **126.8** ± 12.22 |

Table 1: Performance on synthetic and real world datasets with target confidence $1 - \alpha = 0.9$. Methods that are *invalid* (coverage below 90%) are greyed out. CopulaCPTS achieves high level of calibration (coverage is close to 90%) while producing more efficient confidence regions (small area).

For model-dependent baselines MC-dropout and BJRNN, we have to train an RNN forecasting model from scratch for each method, which is additional computational cost.

Results in Table 1 show that, like for synthetic datasets, CopulaCPTS is both more calibrated and efficient compared to baseline models for real world datasets. For the trajectory prediction task, learning the copula results in a 77% sharper confidence region while still remaining valid for the 90% confidence interval. We visualize two samples from each dataset in Figure 3. The importance of efficiency in these scenarios is clear - the confidence regions need to be narrow enough for them to be useful for decision making. Given the same underlying prediction model, we can see that CopulaCPTS produces a much sharper region while still remaining valid.

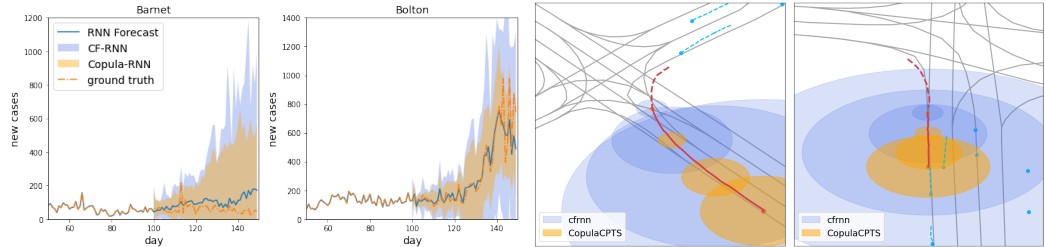

Figure 3: Illustrations of 90% confidence regions given by CF-RNN (blue) and CopulaCPTS (orange) on two real-world datasets. For COVID-19 forecast (left 2) we see that CopulaCPTS produces sharper yet covering confidence regions. For Argoverse (right 2) we illustrate regions at time steps 1, 10, 20, and 30. Note that the confidence region produced CF-RNN is uninformatively large, as it covers all the lanes. Overall, CopulaCPTS is able to produce much more efficient confidence regions while maintaining valid coverage. These examples also illustrate the importance of efficiency.

**Comparison of models at different horizon lengths.** CopulaCPTS is an algorithm designed to produce calibrated and efficient confidence regions for multi-step time series. When the prediction horizon is long, CopulaCPTS's advantage is more pronounced. Figure 4 shows performance comparison across increasing time horizons on the particle dataset. Additional experiment results can be found in Table 3 of Appendix C. CopulaCPTS achieves a 30% decrease in area at 20 time steps compared to CF-RNN, the best performing baseline; the decrease is above 50% at 25. This experiment shows the significant improvement of using copula on modeling the joint distribution of future time steps.

**CopulaCPTS for the Autoregressive prediction setting.** We test the autoregressive setting (section 4.3) on the COVID-19 dataset. We train an RNN model with $k = 7$ and use it to autoregressively forecast the next 14 steps. Table 2 compares the performance of re-estimating the copula for each 7-step forecasts versus using a fixed copula calibrated using the first 7 steps. We compare the model to a 14-step joint forecaster using CopulaCPTS as well.

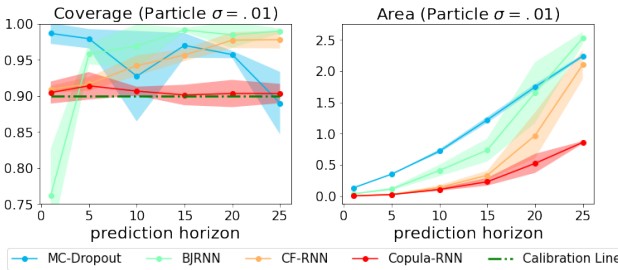

| Method | Coverage | Area |
|---|---|---|
| AR re-estimate | **90.7** | **377.9** |
| AR fixed | 88.2 | 340.5 |
| Joint | 90.8 | 429.4 |

Table 2: Performance of autoregressive (AR) CopulaCPTS. Re-estimating copula gives us valid confidence region over time and is more efficient than joint CopulaCPTS forecast.

Figure 4: CopulaCPTS remains more calibrated and efficient than baselines over increasing forecast horizons.

## 6 CONCLUSION AND DISCUSSION

In this paper we present CopulaCPTS, a conformal prediction algorithm for multidimensional and multi-step time-series prediction. CopulaCPTS significantly improves calibration and efficiency of multi-step conformal confidence intervals by incorporating copulas to model the joint distribution of multiple timesteps. We prove that CopulaCPTS has finite sample validity guarantee. In our experiments we show that the algorithm outperforms state-of-the-art models on all 4 benchmark datasets and on varying prediction horizons. On the flip side, CopulaCPTS assumes that the copula estimation is accurate and we have calibration data for the entirety of the prediction horizon, even in the autoregressive case. We leave it to future work to relax these assumptions (for example by adjusting $\mathbf{u}^*$ online given prediction errors as in Gibbs & Candes (2021)) to make CopulaCPTS applicable to the online setting where distribution shift is present.

## REPRODUCIBILITY STATEMENT

All experiments included in this paper are repeated over 3 runs to account for randomness in neural network training. We report the mean and the standard deviation of the experiment results in our tables. We have included code for constructing the synthetic datasets, and source code for our model implementation, experiments, and visualizations in the supplementary material. For real-world datasets, we refer readers to Appendix C for detailed descriptions of how to obtain and preprocess the data.

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

APPENDIX

## A    PROOF OF PROPOSITION 1

**Proposition 1** (Validity of CopulaCPTS). *The confidence region provided by CopulaCPTS (algorithm 1) a valid. i.e.* $\mathbb{P}[\,\forall h \in \{1, \ldots, k\},\ \mathbf{y}_{t+h} \in \Gamma_h^{1-\alpha}\,] \geq 1 - \alpha$.

*Proof.* We estimated $\mathbf{u}^* = (\hat{F}_1(s_1^*), \ldots, \hat{F}_t(s_k^*))$ such that

$$C_{\text{empirical}}(\mathbf{u}^*) = \mathbb{E}_u \prod_{t=1}^{k} \mathbb{1}_{\mathbf{u}_t^* < \mathbf{u}_t} \geq 1 - \alpha \tag{11}$$

Let $(\mathbf{x}_{1:t}^{n+1}, \mathbf{y}_{t+1:t+h}^{n+1}) \sim \mathbf{X} \times \mathbf{Y}$ be a new data point. Denote $\hat{\mathbf{y}} = \hat{f}(\mathbf{x}_{1:t}^{n+1})$, the forecast given by the trained model. Because the $\hat{F}_h$ functions and $C_{\text{empirical}}$ are monotonously increasing, we have:

$$\begin{aligned} P[\,\forall h \in \{1, \ldots, k\},\ \mathbf{y}_{t+h} \in \Gamma_h^{1-\alpha}\,] &= C(\hat{F}_1(s_1^{n+1}), \ldots, \hat{F}_k(s_k^{n+1})) \\ &\geq C(\hat{F}_1(s_1^*), \ldots, \hat{F}_k(s_k^*)) \\ &\geq 1 - \alpha \end{aligned}$$

$\square$

## B    ADDITIONAL ALGORITHM DETAILS

### B.1    UPPER AND LOWER BOUNDS FOR COPULAS

To provide a better understanding of the properties of Copulas, consider the Frechet-Hoeffding Bounds (Theorem 2). In fact, the Frechet-Hoeffding upper- and lower- bounds are both copulas. The lower bound is percisely the Bonferroni correction used in Stankevičiūtė et al. (2021) - therefore by estimating the copula more precisely instead of using a lower bound, we have a guaranteed efficiency improvement for the confidence region.

**Theorem 2** (The Frechet-Hoeffding Bounds). *Consider a copula* $C(u_1, \ldots, u_k)$. *Then*

$$\max\left\{1 - k + \sum_{i=1}^{k} u_i, 0\right\} \leq C(u_1, \ldots, u_k) \leq \min\{u_1, \ldots, u_k\}$$

### B.2    NUMERICAL OPTIMIZATION WITH SGD FOR SEARCH

The empirical inverse CDF modeled the same way as Equation 6 is the same as the empirical quantile estimation (Equation 2).

$$\hat{F}^{-1}(p) := \inf\{s : (\frac{1}{|D_{cal}|} \sum_{z^i \in D_{cal}} \mathbb{1}_{A(z, \hat{f}) \leq s}) \geq p\} \tag{12}$$

We find the optimal $s_h^*$ in Equation 10 and Algorithm 1 by minimizing the following loss:

$$\mathcal{L}(s_1, \ldots, s_k) = \frac{1}{n-m} \sum_{i=m+1}^{n} \prod_{h=1}^{k} \mathbb{1}\left[\mathbf{u}_h^i < \hat{F}_h^{-1}(s_h)\right] - (1 - \alpha)$$

We use the Adam optimizer and optimize for 500 steps. An alternative and faster way of search is to assume $s_1 = s_2 = \cdots = s_k$ which simplifies the search from exponential time to constant time.

### B.3 Autoregressive prediction

In the context of this paper to forecast autoregressively is given input $\mathbf{x}_{1:t}$ and a $k$ step forecasting model $\hat{f}$, perform prediction

$$\hat{\mathbf{y}}_{t+1:t+k} = \hat{f}(\mathbf{x}_{1:t})$$
$$\hat{\mathbf{y}}_{t+2:t+k+1} = \hat{f}(\mathbf{x}_{2:t}, \hat{\mathbf{y}}_{t+1})$$
$$\dots$$

until all $k'$ time steps are predicted.

We now provide a toy scenario to illustrate when re-estimating the copula is necessary and improves validity. Consider a time series of three time steps $t_0, t_1, t_2$. The two scenarios are illustrated in Figure 5. In both scenarios the mean and variance of all timeseps are 0 and 1 respectively. In scenario (a), $t_0 = t_1$ and hence their covariance is 1. The copula estimated on $t_0$ and $t_1$ is $C_{0:1}(F(t_0), F(t_1)) = F(t_0) = F(t_1)$. This copula will significantly underestimate the confidence region of $t_2$ where its covariance with $t_1$ is $-1$. In fact the coverage of $C_{0:1}(F_1(t_1), F_2(t_2)) = 0.74$. On the other hand, (b) illustrates a scenario where the copula for any 2 consecutive time series remain the same $C_0 = C_1$. In this case, applying $C_0$ directly to forecast $C_1$ achieves precisely 90% coverage.

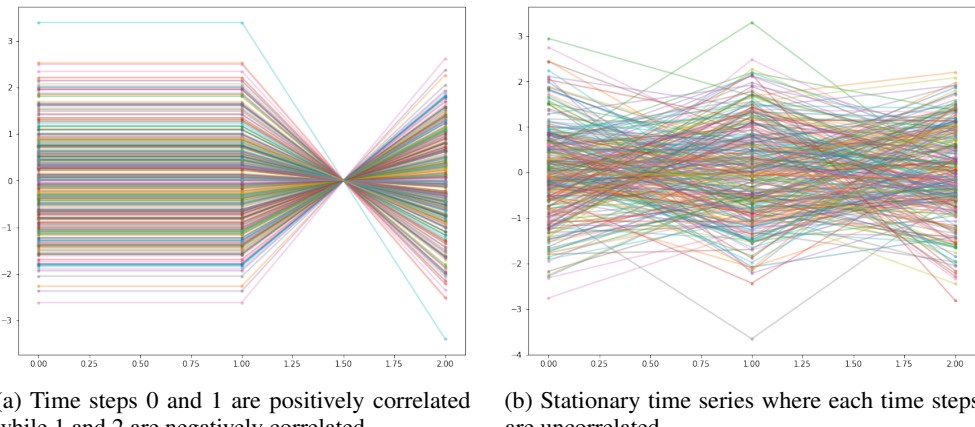

(a) Time steps 0 and 1 are positively correlated while 1 and 2 are negatively correlated

(b) Stationary time series where each time steps are uncorrelated

Figure 5: Two scenarios to illustrate the autoregressive case

## C Experiment Details and additional results

### C.1 Underlying forecasting models

**Particle Dataset**   The underlying forecasting model for the particle experiments is an 1-layer LSTM network with embedding size $= 24$. The hidden state is then passed through a linear network to forecast the timesteps concurrently (output has dimension $k \times d_y$). We train the model for 150 epochs with batch size 128. Hyperparameters of the network are selected through a model search by performance on a 5-fold cross validation split of the dataset. The architecture and hyperparameters are shared for all baselines and CopulaCPTS in Table 1.

**Drone**   For the drone trajectory forecasting task, we use he same LSTM forecasting network as the particle dataset, but with hidden size increased to 128. We train the model for 500 epochs with batch size 128. The same architecture and hyperparameters are shared for all baselines and CopulaCPTS reported in Table 1.

**Covid-19**   The base forecasting model for Covid-19 dataset is the same as the synthetic datasets, with hidden size $= 128$ and were trained for 150 epochs with batch size 128. The same architecture and hyperparameters are shared for all baselines and CopulaCPTS reported in Table 1.

**Argoverse**   As highlighted in the main text, we utilize a state-of-the-art prediction algorithm LaneGCN (Liang et al., 2020) as the underlying forcaster model for CF-RNN and Copula-RNN. We refer the readers to their paper and code base for model details. The architecture of the RNN network used for MC-Dropout and BJRNN is an Encoder-Decoder network. Both the encode and decoder contains a LSTM layer with encoding size 8 and hidden size 16. We chose this architecture because the is part of the official Argoverse baselines (`https://github.com/jagjeet-singh/argoverse-forecasting`) and demonstrates competitive performance.

## C.2   COVID-19 DATASET

The COVID-19 dataset is downloaded directly from the official UK government website `https://coronavirus.data.gov.uk/details/download` by selecting *region* for area type and *newCasesByPublishDate* for metric. There are in total 380 regions and over 500 days of data, depending when it is downloaded. We selected 150-day time series from the collection to construct our dataset.

## C.3   CALIBRATION AND EFFICIENCY CHART FOR COVID-19

Figure 6 shows comparison of calibration and efficiency for the daily new COVID 19 cases forecasting.

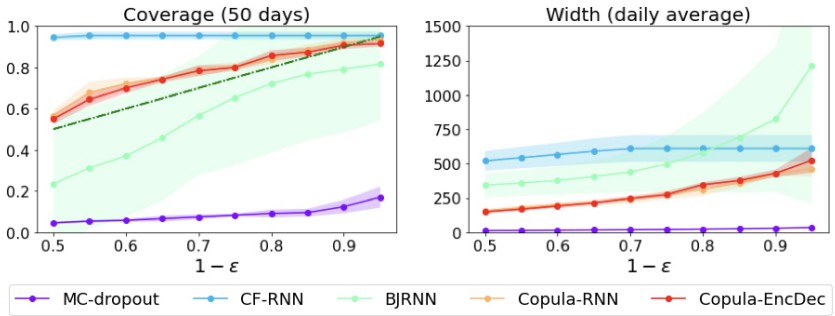

Figure 6: Calibration and efficiency comparison on different $\epsilon$ level for COVID-19 Daily Forecasts. The copula methods (orange and red lines) are more calibrated (coinciding with the green doted line) and sharp (low width) compared to baselines.

To see if the daily fluctuation due to testing behaviour disrupts other method, we also ran the same experiment on weekly aggregated new cases forecast. We take 14 weeks of data as input and output forecasts for the next 6 weeks. The results are illustrated in Figure 7. The weekly forecasting scenario gives us similar insights as the daily forecasts.

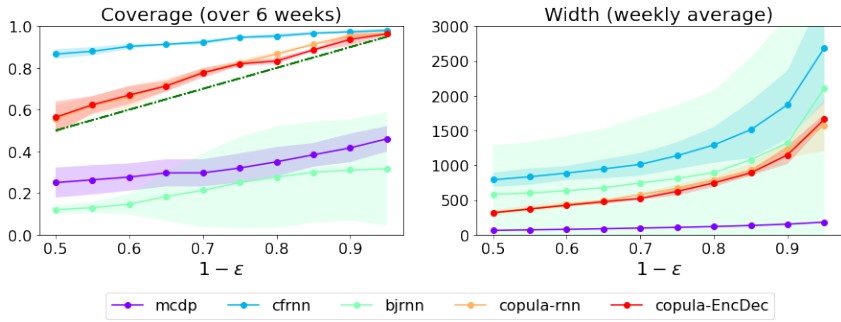

Figure 7: Covid Weekly Forecasts

## C.4 ARGOVERSE

The Argoverse autonomous vehicle dataset contains 205,942 samples, consisting of diverse driving scenarios from Miami and Pittsburgh. The data can be downloaded from the official Argoverse dataset website. We split 90/10 into a training set and validation set of size 185,348 and 20,594 respectively. The official validation set of size 39,472 is used for testing and reporting performance. We preprocess the scenes to filter out incomplete trajectories and cap the number of vehicles modeled to 60. If there are less than 60 cars in the scenario, we insert dummy cars into them to achieve consistent car numbers. For map information, we only include center lanes with lane directions as features. Similar to vehicles, we introduce dummy lane nodes into each scene to make lane numbers consistently equal to 650.

## C.5 ADDITIONAL EXPERIMENT RESULTS

We present in figure 8 and 9 some qualitative results for uncertainty estimation.

To test how the effects of copulaCPTS compare with baseline on other base forecasters, we also include an encoder-decoder architecture with the same embedding size as the RNN models introduced in appendix C.1 for each dataset. The results are presented in Table 3. We omit these results in the main text because we found that they do did not bring significant improvement to time series forecasting UQ.

Table 4 compares model performance compared across different prediction horizons. We show that the advantage of our method is more pronounced for longer horizon forecasts.

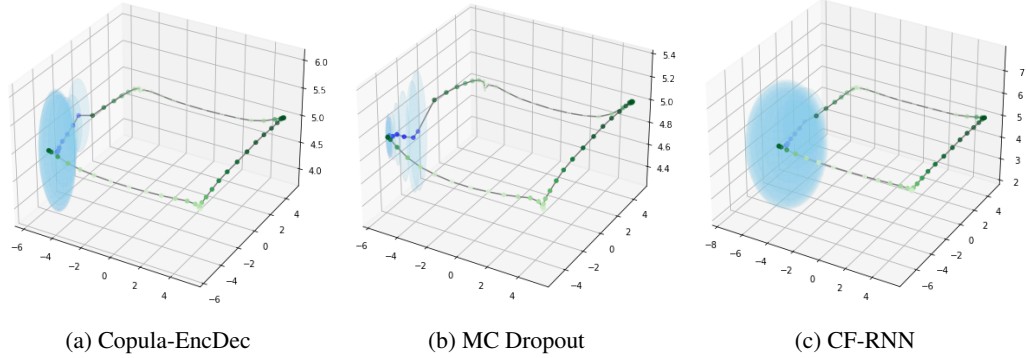

(a) Copula-EncDec        (b) MC Dropout        (c) CF-RNN

Figure 8: 99% Confidence region produced by three methods for the drone dataset. Copula methods (a) produces a more consistent, expanding cone of uncertainty compared to MC-Dropout (b) sharper one compared to CF-RNN (c).

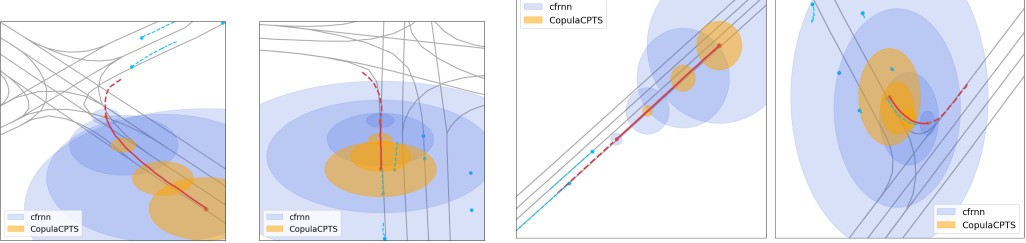

Figure 9: Illustrations for confidence regions given by CF-RNN (blue) and CopulaCPTS (orange) in at time steps 0, 10, 20, and 30. Note that in order to achieve 90% coverage, the regions are larger than needed, especially in straight-lane cases like the middle two. Using copulas to couple together time steps results in a much smaller region, while achieving similarly good coverage.

| Particle Simulation ($\sigma = .01$) | | | | |
|---|---|---|---|---|
| | Coverage (90%) | Area (90%) | Coverage (99%) | Area (99%) |
| MC-dropout | 63.9 $\pm$ 16.1 | 0.81 $\pm$ 0.05 | 73.1 $\pm$ 14.2 | 1.14 $\pm$ 0.07 |
| BJRNN | 98.9 $\pm$ 0.2 | 2.24 $\pm$ 0.59 | 100 $\pm$ 0.0 | 3.75 $\pm$ 0.71 |
| CF-RNN | 97.0 $\pm$ 2.3 | 3.13 $\pm$ 3.24 | 100 $\pm$ 0.21 | 4.34 $\pm$ 1.19 |
| Copula-RNN | 91.5 $\pm$ 2.1 | **1.06** $\pm$ 0.36 | 97.4 $\pm$ 2.4 | 4.47 $\pm$ 1.48 |
| Copula-EncDec | **90.2** $\pm$ 2.3 | 1.19 $\pm$ 0.17 | **99.6** $\pm$ 0.7 | **1.83** $\pm$ 0.09 |

| Particle Simulation ($\sigma = .05$) | | | | |
|---|---|---|---|---|
| | Coverage (90%) | Area (90%) | Coverage (99%) | Area (99%) |
| MC-dropout | 16.1 $\pm$ 4.3 | 0.79 $\pm$ 0.02 | 33.9 $\pm$ 5.1 | 2.12 $\pm$ 0.03 |
| BJRNN | 100.0 $\pm$ 0.0 | 12.13 $\pm$ 0.39 | 100.0 $\pm$ 0.0 | 15.43 $\pm$ 0.85 |
| CF-RNN | 94.5 $\pm$ 1.5 | 5.79 $\pm$ 0.51 | 99.8 $\pm$ 2.2 | 19.21 $\pm$ 8.19 |
| Copula-RNN | **90.3** $\pm$ 0.7 | 4.50 $\pm$ 0.07 | **99.1** $\pm$ 0.8 | 12.82 $\pm$ 3.98 |
| Copula-EncDec | 91.4 $\pm$ 1.1 | **4.40** $\pm$ 0.15 | 98.7 $\pm$ 0.1 | **9.31** $\pm$ 1.97 |

| Drone Simulation ($\sigma = .02$) | | | | |
|---|---|---|---|---|
| | Coverage (90%) | Area (90%) | Coverage (99%) | Area (99%) |
| MC-dropout | 84.5 $\pm$ 10.8 | 9.64 $\pm$ 2.13 | 90.0 $\pm$ 7.8 | 16.02 $\pm$ 3.62 |
| BJRNN | 90.8 $\pm$ 2.8 | 49.57 $\pm$ 3.77 | 100.0 $\pm$ 4.0 | 65.77 $\pm$ 4.56 |
| CF-RNN | 91.6 $\pm$ 9.2 | 32.18 $\pm$ 13.66 | 100.0 $\pm$ 0.0 | 36.79 $\pm$ 14.03 |
| CF-EncDec | 100.0 $\pm$ 0.0 | 21.83 $\pm$ 26.29 | 100.0 $\pm$ 0.0 | 25.03 $\pm$ 12.53 |
| Copula-RNN | **90.0** $\pm$ 1.5 | **16.52** $\pm$ 15.08 | **98.5** $\pm$ 0.5 | **21.48** $\pm$ 8.91 |

| COVID-19 Daily Cases Dataset | | | | |
|---|---|---|---|---|
| | Coverage (90%) | Area (90%) | Coverage (99%) | Area (99%) |
| MC-dropout | 19.1 $\pm$ 5.1 | 34.14 $\pm$ 0.84 | 100.0 $\pm$ 0.0 | 1106.57 $\pm$ 25.41 |
| BJRNN | 79.2 $\pm$ 30.8 | 823.3 $\pm$ 529.7 | 85.7 $\pm$ 27.5 | 149187. $\pm$ 51044. |
| CF-RNN | 95.4 $\pm$ 1.9 | 610.2 $\pm$ 96.0 | 100.0 $\pm$ 0.0 | 121435. $\pm$ 26495. |
| CF-EncDec | 91.7 $\pm$ 1.4 | 570.3 $\pm$ 22.1 | 100.0 $\pm$ 0.0 | 108130. $\pm$ 10889. |
| Copula-RNN | 92.1 $\pm$ 1.0 | **429.0** $\pm$ 15.1 | 100.0 $\pm$ 0.0 | 88962. $\pm$ 9643. |
| Copula-EncDec | **90.8** $\pm$ 0.3 | 429.4 $\pm$ 27.9 | 100.0 $\pm$ 0.0 | **60852.** $\pm$ 12263. |

| Argoverse Trajectory Prediction Dataset | | | | |
|---|---|---|---|---|
| | Coverage (90%) | Area (90%) | Coverage (99%) | Area (99%) |
| MC-dropout | 27.9 $\pm$ 3.1 | 127.6 $\pm$ 20.9 | 31.5 $\pm$ 3.9 | 242.1 $\pm$ 54.0 |
| BJRNN | 92.6 $\pm$ 9.2 | 880.8 $\pm$ 156.2 | 100.0 $\pm$ 0.0 | 3402.8 $\pm$ 268. |
| CF-LaneGCN | 98.8 $\pm$ 1.9 | 396.9 $\pm$ 18.67 | 100. $\pm$ 0.2 | 607.2 $\pm$ 8.67 |
| Copula-LaneGCN | **90.4** $\pm$ 0.3 | **126.8** $\pm$ 12.22 | **99.1** $\pm$ 0.4 | **324.1** $\pm$ 42.22 |

Table 3: Additional results. Copula methods achieves high level of calibration while producing sharper predictions regions. The sharpness gain is even more pronounced in at higher confidence levels (99%), where we want the prediction region to be useful while remaining valid.

| Method | 1 Step Coverage | 1 Step Area | 5 Steps Coverage | 5 Steps Area | 15 Steps Coverage | 15 Steps Area |
|---|---|---|---|---|---|---|
| MC-Dropout | 97.8 ± 2.0 | 0.4 ± 0.04 | 88.0 ± 7.0 | 0.69 ± 0.25 | 52.3 ± 1.4 | 0.94 ± 0.2 |
| BJRNN | 45.3 ± 39.4 | 0.27 ± 0.18 | 97.7 ± 2.1 | 2.69 ± 1.79 | 95.5 ± 2.8 | 19.99 ± 4.83 |
| CF-RNN | 100.0 ± 0.0 | **0.01** ± 0.01 | 77.8 ± 19.2 | 0.8 ± 0.64 | 66.7 ± 0.0 | 18.82 ± 3.73 |
| CF-EncDec | 89.9 ± 19.2 | **0.01** ± 0.01 | 100.0 ± 0.0 | 0.75 ± 0.99 | 88.9 ± 19.2 | 13.07 ± 16.1 |
| Copula-RNN | 90.1 ± 0.2 | **0.01** ± 0.01 | 89.8 ± 0.6 | **0.54** ± 0.45 | **90.1** ± 1.2 | 8.25 ± 3.44 |
| Copula-EncDec | **90.0** ± 0.3 | **0.01** ± 0.0 | **90.3** ± 0.6 | 0.67 ± 1.01 | 90.5 ± 0.5 | **7.13** ± 9.5 |

Table 4: Performance comparison across different horizons at 90% confidence level on the drone simulation dataset. The improvement on efficiency is more pronounced when the horizon is longer.

### C.6 STUDY ON $\alpha_h$ SEARCH

Figure 10 shows the $\alpha_h$ values for each $1 - \alpha_h = \hat{F}_h(s_h^*)$ used in Copula CPTS as outlined in line 15 of Algorithm 1. We present $\alpha_h$ values searched using two methods of searching, with dichotomy search for a constant $\alpha$ value for the horizon as in Messoudi et al. (2021), and by stochastic gradient descent as outlined in section 4.2.

The $\alpha_h$ values are an indicator of how interrelated the uncertainty between each time step are: Bonferroni Correction used in Stankevičiūtė et al. (2021) (grey dotted line in Figure 10) assumes that the time steps are independent, with CopulaCPTS we have lower $1 - \alpha_h$ levels while having valid coverage (blue and orange lines in Figure 10). This shows that the uncertainty of the time steps are not independent, and we are able to utilize this dependency to shrink the confidence region and still maintaining the coverage guarantee.

Table 5 shows that there are no significant difference between coverage and area performance for the two search methods with in the scope of datasets we study in this paper. However, we want to highlight that SGD search is $O(n)$ complexity to optimization steps, regardless to the prediction horizon. SGD also allows for varying $\alpha_h$ which might be useful in some settings, for example capturing uncertainty spikes for some time steps as seen in the COVID-19 dataset of Figure 10. Dichotomy search, on the other hand, is $O(nlog(n))$ complexity to the search space depends on granularity, and will be $O(knlog(kn))$ if we want to search for varying $\alpha_h$.

| Dataset | Coverage (90%) Fixed $\alpha_h$ | Coverage (90%) Varying $\alpha_h$ | Area Fixed $\alpha_h$ | Area Varying $\alpha_h$ |
|---|---|---|---|---|
| Particle ($\sigma = .01$) | 91.7 ± 1.9 | 91.5 ± 2.1 | 1.13 ± 0.45 | 1.06 ± 0.36 |
| Particle ($\sigma = .05$) | 92.1 ± 1.3 | 90.3 ± 0.7 | 4.89 ± 0.05 | 4.50 ± 0.07 |
| Drone | 90.3 ± 0.5 | 90.0 ± 1.5 | 15.92 ± 1.98 | 16.52 ± 7.08 |
| Covid-19 | 92.9 ± 0.1 | 92.1 ± 1.0 | 498.44 ± 6.36 | 429.0 ± 15.1 |
| Argoverse | 90.2 ± 0.1 | 90.4 ± 0.3 | 117.1 ± 7.3 | 126.8 ± 12.2 |

Table 5: Coverage and area comparison between stochastic search for fixed $\alpha_h$ and SGD for Varying $\alpha_h$. We do not see significant difference between the performance of two.

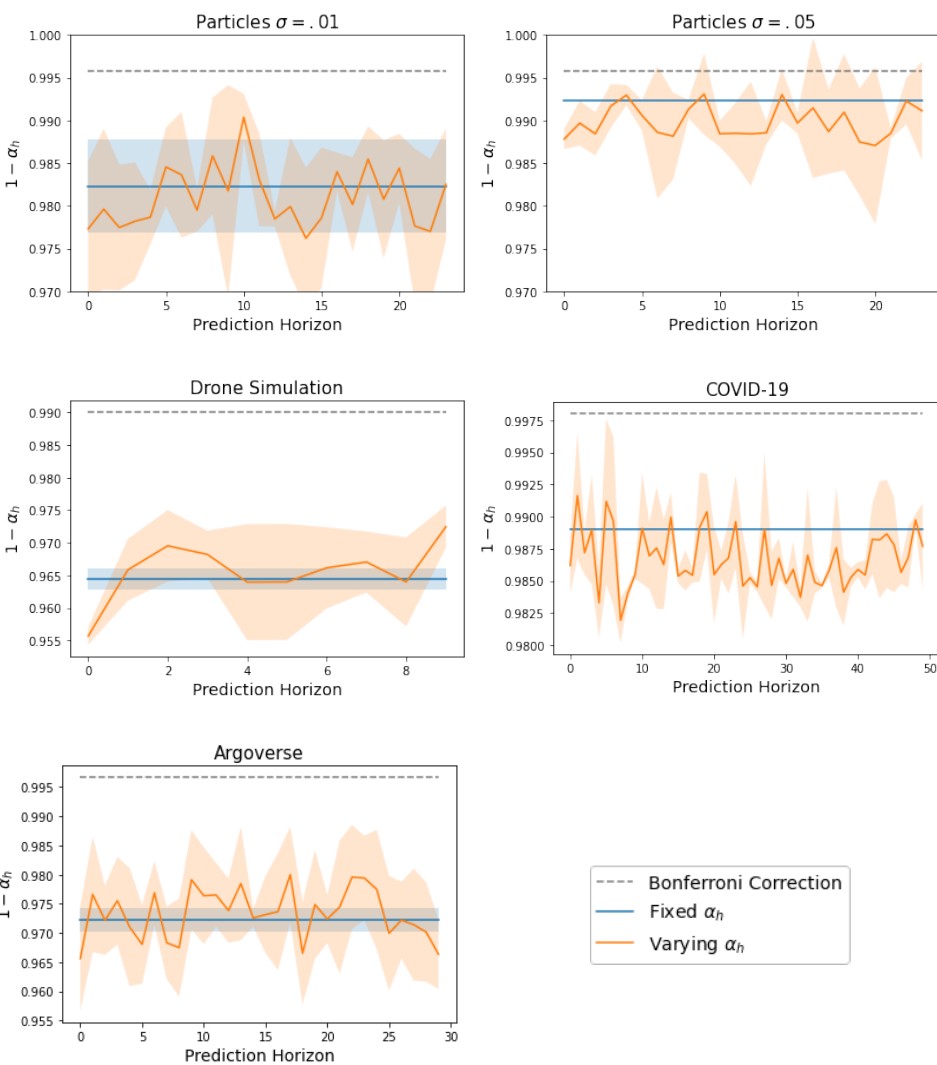

Figure 10: Comparison between dichotomy search for fixed $\alpha_h$ values (blue) and stochastic gradient search for varying $\alpha_h$ (blue) through timesteps. Shaded regions are the standard deviation of the values over 3 runs.

### C.7 COMPARISON TO ADDITIONAL BASELINES

We include a comparison to two additional simple UQ baselines on the particle simulation dataset.

**L2-Conformal.** L2-Conformal uses the same underlying RNN forecaster as CF-RNN and Copula RNN. We use that nonconformity score of the vector norm of all timesteps concatenated together $\|\hat{\mathbf{y}}_{t+1:t+k} - \mathbf{y}_{t+1:t+k}\|$ to perform ICP. As there are no analytic way to represent a $k \times d_y$-dimensional uncertainty region on 2-D space, we calculate the area and plot the region for L2 Conformal baseline with the maximum deviation at each timestep such that the vector norm still stays within range.

**Direct Gaussian.** Direct Gaussian use the same model architecture and training hyperparameters, with the addition of a linear layer that outputs the variance for each timestep, and is optimized using negative log loss, a proper scoring rule for probabilistic forecasting. We obtain the area by analytically calculating the 90% confidence interval for each variable.

Results in Table 6 shows that L2-conformal produces inefficient confidence area, and directly outputting variance under-covers test data. These results align with previous findings and motivates our method, which is both more calibrated and sharper compared to these baselines. We show a visualization in Figure 11 to illustrate the different properties of the methods qualitatively.

| Method | Particle ($\sigma = .01$) | | Particle ($\sigma = .05$) | |
|---|---|---|---|---|
| | Coverage (90%) | Area $\downarrow$ | Coverage (90%) | Area $\downarrow$ |
| L2-Conformal | $88.5 \pm 0.4$ | $7.21 \pm 0.35$ | $89.7 \pm 0.6$ | $7.21 \pm 0.35$ |
| Direct Gaussian | $11.9 \pm 0.09$ | $0.07 \pm 0.31$ | $0.0 \pm 0.0$ | $0.08 \pm 0.02$ |
| CF-RNN | $97.0 \pm 2.3$ | $3.13 \pm 3.24$ | $97.0 \pm 2.3$ | $5.79 \pm 0.51$ |
| Copula-RNN | $\mathbf{91.5} \pm 2.1$ | $\mathbf{1.06} \pm 0.36$ | $\mathbf{90.3} \pm 0.7$ | $\mathbf{4.50} \pm 0.07$ |

Table 6: Comparison with two additional baselines on the particle dataset.

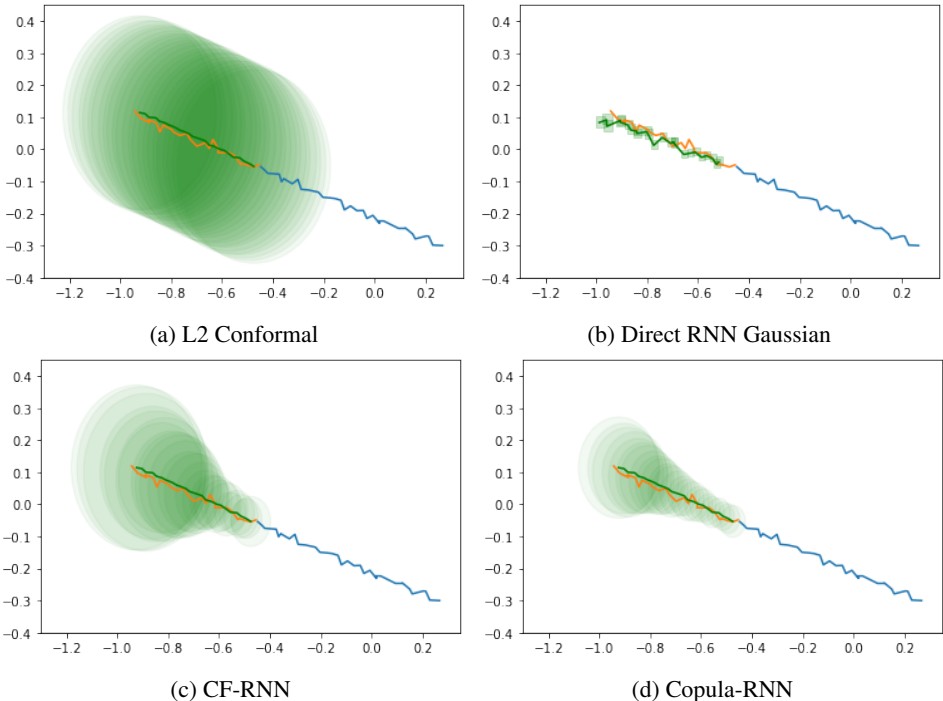

(a) L2 Conformal

(b) Direct RNN Gaussian

(c) CF-RNN

(d) Copula-RNN

Figure 11: Visualization of on a sample from the Particle dataset's test set.

