# OpenReview forum: "Copula Conformal Prediction for Multi-step Time Series Forecasting"
_ICLR.cc/2023/Conference — Submitted to ICLR 2023_

### Official Review · Reviewer_FqUe · 2022-10-21

**Confidence:** 4
**Correctness:** 1
**Technical Novelty And Significance:** 2
**Empirical Novelty And Significance:** 2
**Recommendation:** 3

**Clarity, Quality, Novelty And Reproducibility:**

(clarity) The paper is mostly okay, but some clarification is required.
* Page 4: The citation (Gibbs & Candes, 2021) in the first paragraph is not appropriate; the cited paper attacks online learning, but the submitted paper does not consider the same setup, making readers confusing.
* Page 5: why is (8) true? The Sklar’s theorem only implies the first equality.
* Page 5: epsilon_t below the (8) is not defined.
* Page 7: in “Metrics”, the expectation in the definition of coverage does not make sense to me.
* For the Covid19 dataset, it consists of 380 sequences; how can it be collected in UK regions? It would be better if the paper is self-contained. Is there any chance that the exchangeability assumption can be violated for this dataset?

(Quality) Based on the discussion on weaknesses, the paper can be improved.

(Novelty) I think exploiting copula may be novel once discussed weaknesses are addressed.


**Strength And Weaknesses:**

**Strengths**:

* Highlight the concept on Copula for time-series data
* Extensive experiments on both simulated and interesting real datasets.

**Weaknesses**:
* Proposition 1 is invalid (or clarification is required)
* A simple but important baseline is missing

(Weakness 1) First of all, the ICP validity definition on coverage (Definition 1)  is not correct (unless there are reasons, which are not described). The probability needs to be taken over X, Y, and also a calibration set (e.g., [R1]). Also, this coverage guarantee is not used consistently. In particular, the validity statement in Proposition 1 contains the expectation over y, but to my understanding, the probability here is also taken over y. I guess these misunderstandings might contribute to the incorrect proof of Proposition 1. In particular, (11) assumes u* is given, which needs to be found in Line 15 of Algorithm 1 based on a calibration set. As u* depends on a calibration set, it is a random vector. However, this randomness is not taken into account in the proof as the probability in the proof is also taken only over a new data point. Finally, it is unclear how the exchangeability assumption is used in the proof.

To my understanding, the proof needs to be corrected based on the right coverage definition, and the proposed approach is reevaluated based on the correction.

[R1] https://arxiv.org/pdf/1904.06019.pdf


(Weakness 2) I think the beauty of conformal prediction is that it provides a coverage guarantee for any score function, as also highlighted in the paper. Thus, one straightforward way to handle time-series data is using a score function that aggregates scores of future time steps and simply runs the standard ICP on the aggregated score. For example, based on the paper’s notation, a RNN output and a label are $\hat{\mathbf{y}}, \mathbf{y} \in \mathbb{R}^{k \times d_y}$, respectively, then, a score function can be simply $|| \hat{\mathbf{y}} - \mathbf{y} ||$, which is equivalent to the negative log likelihood of the Gaussian with the mean of $\hat{\mathbf{y}}$ and the identity covariance. This score function can be improved by estimating the diagonal of the covariance matrix (assuming zero correlation) by modifying RNN to estimate the standard deviations of $\hat{\mathbf{y}}$ (I believe this is a quite standard techniques to convert a point estimator to a uncertainty estimator, e.g.,[R2,R3]). Given any of these two score functions, we can run the standard ICP.

I think the limitations of these two simple baselines need to be discussed and empirically evaluated to justify the efficacy of the proposed approach. Moreover, we can use more sophisticated forecasting model (e.g. Bayesian filtering with state evolution [R4,R5]) for a score function along with the standard ICP for the coverage guarantee. What’s the limitation of having a sophisticated score function for time-series data with the standard ICP for the coverage guarantee compared with the proposing new conformal prediction dedicated to time-series data along with a naive score function?

[R2] https://openaccess.thecvf.com/content_CVPR_2019/papers/He_Bounding_Box_Regression_With_Uncertainty_for_Accurate_Object_Detection_CVPR_2019_paper.pdf
[R3] https://arxiv.org/pdf/2001.00106.pdf
[R4] https://arxiv.org/pdf/1805.11122.pdf
[R5] https://arxiv.org/pdf/1605.07148.pdf



**Summary Of The Paper:**

This paper proposes a novel conformal prediction method for time-series data, called Copula Conformal Prediction for multi-step time series forecasting (CopulaCPTS), for uncertainty quantification. In particular, the paper uses the concept of “copula” from statistics; the copula is a joint CDF on data, which fully captures the dependency structure of the data, like in time-series data. The proposed CopulaCPTS theoretically justified its validity (in Proposition 1), and empirically evaluated seven synthetic datasets and two real datasets for its efficacy. The claimed contribution are (1) CopulaCPTS is a general UQ algorithm for any multivariate multi-step forecaster with statistical validity, (2) CopulaCPTS produces sharper uncertainty estimates compared with other approaches, and (3) CopulaCPTS produces valid confidence intervals for time-varying time-series prediction.

**Summary Of The Review:**

The core contribution of the paper is the coverage guarantee of the proposed approach, but to my understanding, the proof is not valid; so I would vote for rejection. However, I’m willing to adjust my understanding after discussion.

========== after rebuttal

Thanks for the clarification and additional experiments. I believe that exploiting the concept of copula is interesting, but (1) the paper relies on a strong assumption (i.e., an accurate Copula is given, which is even not explicitly mentioned in the paper or written in a confusing way), (2) my main concerns were not addressed during the rebuttal, and (3) I think the paper is not ready to publish based on the current manuscript status; so, I maintain my score.


The following includes the remaining main concerns.

* The probability in coverage is taken over a calibration set. Note that the probability in the full CP coverage is taken over a training set (aka a proper training set in CP), a calibration set (aka a training set in CP), and a new sample. The probability in inductive CP (ICP) coverage is taken over a calibration set and a new sample (where the training set is used for learning a score function). The probability in training-conditional ICP coverage is taken over a new sample. I’d recommend checking the difference between ICP and training-conditional ICP in [1].
* The above definition is crucial in proving Proposition 1 if the accurate Copula is unknown. If Proposition 1 is proven without this strong assumption, I’m wondering if the calibration set can be reused so Bonferroni correction needs to be used in this step. In the worst case, the rigorous Proposition 1 produces a more conservative algorithm such that a more rigorous proposed approach may not be clearly better than CF-RNN. In other words, by making an assumption that the accurate Copula is known, the proposed algorithm becomes heuristic, and it looks unfair to compare it with a more rigorous approach like CF-RNN — usually rigorous algorithms are more conservative (in achieving a desired coverage) than heuristic ones. In this sense, I think proving Proposition 1 by estimating a Copular from data is required as for the first step.

---

> ### Author Response · Authors · 2022-11-18
> **Response to Reviewer FqUe (part 1)**
>
>
> We would like to sincerely thank Reviewer FqUe for providing these comments. We believe there are some misunderstandings, as we will outline in this response.
>
> ### Response to Weakness 1
> > the ICP validity definition on coverage (Definition 1) is not correct (unless there are reasons, which are not described). The probability needs to be taken over X, Y, and also a calibration set (e.g., [R1])
>
> We respectfully disagree. Our definition of validity follows from seminal works on conformal prediction: see proposition 1 of Vovk (2013) [1] and the introduction section of Lei (2018) [2]. We want to point out that $\mathcal{P}$ in our definition 1 _**is**_ the data distribution for $(X,Y)$. In the regression case, we can represent the training, calibration, and test data as an exchangeable set of data drawn from distribution $\mathcal{P}$. Theorem 1 in [R1] is identical to our formulation. It omits the subscript because the coverage inequality holds for arbitrary data distribution $\mathcal{P}$.
>
> [1] Vovk, Vladimir. "Conditional validity of inductive conformal predictors." Asian conference on machine learning. PMLR, 2012.
> [2] Lei, Jing, Max G’Sell, Alessandro Rinaldo, Ryan J. Tibshirani, and Larry Wasserman. "Distribution-free predictive inference for regression." Journal of the American Statistical Association 113, no. 523, 2018.
>
> > Also, this coverage guarantee is not used consistently. In particular, the validity statement in Proposition 1 contains the expectation over y, but to my understanding, the probability here is also taken over y.
>
> The expectation in proposition 1's statement is a typo - we sincerely apologize for it. The inequality should be on the probability as written in the proof, not on the expectation. The content of the proof (in appendix A) is correct and is what we mean for the coverage guarantee.
>
> We have removed the extraneous expectation notation in the updated version of the paper. We believe Proposition 1 in the paper is now correct.
>
> > In particular, (11) assumes $u^*$ is given, which needs to be found in Line 15 of Algorithm 1 based on a calibration set. As $u^*$ depends on a calibration set, it is a random vector.
>
> We want to highlight that the $u^*$ does not depend on the calibration set, but only on the copula function $C$. Given a copula $C$, the set of value for $u^*$ such that $C(u^*) = 1-\alpha$ is fixed. Sklar's theorem (theorem 1) proves the existence of this $u^*$. The proposition is proven with the perfect $u^*$ here; we assume our search for $u^*$ is accurate give the law of large numbers.
>
> >  Finally, it is unclear how the exchangeability assumption is used in the proof.
>
> The exchangeability assumption is used when estimating $\hat{F}$ functions (equation 6 and 7). Note that equation (6) is an ICP procedure; the equation holds because of the exchangeability assumption and the coverage guarantee that follows (equation 3). CopulaCPTS combines the results of $k$ individual ICP procedures for a cumulative guarantee.
>
> Please let us know if this explanation is clear. If not, what can we add to the proof to clarify?
>
> ### Response to Weakness 2
>
> > Missing baselines:
> > - A score function can be simply be $\| \hat{\mathbf{y}} - \mathbf{y} \|$
> > - by modifying RNN to estimate the standard deviations of $\hat{\mathbf{y}}$
>
>
> We have included the baselines per request in Appendix C.7 of the updated paper. However, we justify our choice of not including them in the main text as follows.
>
> Using L2 norm over the entire forecast horizon as nonconformity score has two drawbacks: (1) L-2 distance does not grant us the desired property of creating a cone of uncertainty for time-series forecasting. (2) Form an application perspective, it is limited in the ability to help aid decisions. In the Covid forecasting scenario, we want to know the highest and lowest number of cases per day with high confidence. For robotics applications, we want to draw bounding boxes around our prediction so we can navigate around with safety. L2 norm does not give direct guidance on the uncertainty for each timestep. (To visualize the uncertainty estimate we assume that for each timestep all error comes from that one step. This results in very inefficient confidence regions.)
>
> Directly using an RNN to estimate the standard deviations is widely know to be prone to over-confidence [3], hence more sophisticated approaches such as the papers you cited are researched. There is also a misunderstanding here: if one directly estimate the variance with an RNN, one make the assumption that the error distribution is Gaussian. Conformal prediction is a distribution-free UQ method and avoids making such assumptions. You wouldn't need CP once you have a Gaussian estimation and can directly obtain confidence regions, as we did in Figure 11(b) in Appendix C.7.
>
> [3] Guo, Chuan, Geoff Pleiss, Yu Sun, and Kilian Q. Weinberger. "On calibration of modern neural networks." In International conference on machine learning. PMLR, 2017.

---

> > ### Comment · Reviewer_FqUe · 2022-11-19
> > **Thanks for the response**
> >
> > I appreciate the authors' response. I read it carefully, but my main concerns still remain.
> >
> > > Our definition of validity follows from seminal works on conformal prediction: see proposition 1 of Vovk (2013) [1] and the introduction section of Lei (2018) [2].
> >
> > The coverage definition of [R1], [1], and [2] are equivalent. But, Definition 1 of the paper is not written in the equivalent way, or at least written in a confusing way. The probability is only taken over a new sample $(X, Y)$, but based on [R1], [1], and [2], it is taken over the new point (X, Y) and calibration samples $(X_1, Y_1), \dots, (X_{n-m}, Y_{n-m})$. I cannot find a reason to drop this crucial information — whether the probability is taken over calibration samples or not is a crucial difference between ICP and training-conditional ICP in [1].
> >
> > If authors think Definition 1 is equivalent to [R1], [1], and [2], please explicitly denote that the probability is also taken over calibration samples.
> >
> >
> > > We want to highlight that the $u^*$ does not depend on the calibration set
> >
> > I’m not sure if it is a typo or not, but it does not make sense to me. $u^*$ is defined in Eq. (10), which depends on $C_\text{empirical}$, and $C_\text{empirical}$ depends on the calibration set as written in Eq. (9).
> >
> > For sure, if $C$ is magically given, instead of $C_\text{empirical}$, based on the infinite number of calibration sets, $u^*$ does not depend on the calibration set; but this is not conformal prediction — conformal prediction provides a finite sample guarantee, which makes it very practical, unlike a traditional asymptotical guarantee.
> >
> > The proof needs to consider the randomness from the calibration set. I think an asymptotic guarantee is okay, but the paper still needs to change notations. More importantly, I’m not sure if it is acceptable to call the proposed approach conformal prediction — to my understanding, better to call it something else.
> >
> > > The exchangeability assumption is used when estimating $\hat{F}$ functions (equation 6 and 7). Note that equation (6) is an ICP procedure; the equation holds because of the exchangeability assumption and the coverage guarantee that follows (equation 3). CopulaCPTS combines the results of $k$ individual ICP procedures for a cumulative guarantee.
> >
> > I think the exchangeability assumption is also explicitly used in proving Proposition 1, as $u^*$ depends on the calibration set.
> >
> > Also, as mentioned, Eq. (6) and Eq. (7) depend on $k$ ICP results. But, my concern is that these $k$ ICP uses the same calibration set multiple times. From this, making the cumulative guarantee is not clear. In particular, this issue is related to Bonferroni correction as used in CF-RNN, but it is not clear how this paper can avoid Bonferroni correction (or similar other corrections). To my current understanding, the paper ignores any correction but simply allows the reuse of a calibration set, which is not rigorous. If correctly fixed, I’m wondering whether the proposed approach is still better than CF-RNN.
> >
> >
> >
> > > We have included the baselines per request in Appendix C.7 of the updated paper. However, we justify our choice of not including them in the main text as follows.
> >
> > Thanks for adding additional baselines! For the “Direct Gaussian”, what I meant is use this as a score function, and simply use ICP, as the paper also did for “L2-Conformal” for the additional baseline. In this way, the baseline can achieve a desired coverage guarantee in a very simple way.
> >
> > The main reason for the importance of this kind of baselines is that I currently believe that there is a very simple way to address the paper’s problem, by combining well-developed Baysian forecasting along with a simple application of ICP. In particular, Figure 11(b) demonstrates that if the intervals are correctly chosen based on ICP, “Direct Gaussian” may provide a smaller interval size than the proposed approach, while satisfying the coverage guarantee simply due to ICP.
> >
> >
> > > There is also a misunderstanding here: if one directly estimate the variance with an RNN, one make the assumption that the error distribution is Gaussian. Conformal prediction is a distribution-free UQ method and avoids making such assumptions.
> >
> > As mentioned above, I did not suggest making an assumption of the error distribution. Instead, use the Guassian as a score function for ICP as ICP works with any score function, as the paper did for “L2-Conformal” for the additional baseline.
> >
> >
> > Final remark: as described and based on the current paper draft, my main concerns still remain, so I could not change my score. I hope my understanding is correct and the feedback is useful.

---

> > > ### Author Response · Authors · 2022-11-19
> > > **Clarifying some confusions**
> > >
> > > We appreciate your careful review and reply.  (this response has been edited on Nov 20.)
> > >
> > > >  please explicitly denote that the probability is also taken over calibration samples.
> > >
> > > The probability is not taken over the calibration set. Please refer to Proposition 1 of [1]. The precise wording is
> > >
> > > "the probability of error $Y_{l+1} \notin \Gamma^{\epsilon}(\ldots)$ does not exceed $\epsilon$."
> > >
> > > which is a probability for the test sample $Y_{l+1}$.
> > >
> > > If you are referring to training data in Proposition 1 of [1] that is $\Gamma^{1-\alpha}(z_1, \ldots, z_l, x)$, we want to point out that the training set is incorporated in ICP to train the prediction function, and hence are often omitted in the $\Gamma^{1-\alpha}$ function expressions. Theorem 1 in [R1] is an example (they use the $\hat{C}_n(x)$ expression).
> > >
> > > [1] Vovk, Vladimir. "Conditional validity of inductive conformal predictors." Asian conference on machine learning. PMLR, 2012.
> > >
> > > > The proof needs to consider the randomness from the calibration set. I think an asymptotic guarantee is okay, but the paper still needs to change notations.
> > >
> > > In this paper we assume that the empirical copula estimated on the calibration set is accurate. We highlighted this assumption in section 6 of the original paper and left relaxation of this assumption for future work (which may lead to an asymptotic guarantee as you said). Thank you for your feedback - we will made it more clear in section 4.2 above equation (10) and in the proof of Proposition 1 in appendix A. We hope this assumption can be obvious to the readers.
> > >
> > > As for naming traditions, the name Conformal Prediction more so refer to the algorithm and technique rather than the guarantee in particular. In conformal prediction literature this name has been used for methods that don't have finite sample validity guarantees [4].
> > >
> > > [4] Gibbs, Isaac, and Emmanuel Candes. "Adaptive conformal inference under distribution shift." Advances in Neural Information Processing Systems 34 (2021).
> > >
> > > > Bonferroni correction, reusing calibration dataset.
> > >
> > > We use the calibration set exactly the same as CF-RNN [2], by treating each forecasting timestep $h$ as a separate task. By validity of ICP, each of these $k$ estimations are individually valid.
> > >
> > > Bonferroni correction, and hence CF-RNN, allows us to make a cumulative guarantee because of Boole’s Inequality:
> > >
> > > $\mathbf{P}[ \bigcup_{i=1}^k (p_i\leq \frac{\alpha}{k}) \] \leq \sum_{i=1}^k (\mathbf{P}[p_i \leq \frac{\alpha}{k} ] )= k \cdot \frac{\alpha}{k} = \alpha$
> > >
> > > In the second paragraph of Section 5.1, we noted that Bonferroni correction is an upper bound of Copula functions. The idea of our paper is that if we can find an accurate Copula then CopulaCPTS is theoretically guaranteed to have smaller regions than CF-RNN.
> > >
> > > Learning the copula with the calibration set does not constitute reusing it for calibration, as there is no conformal prediction after the $k$ ICP for each forecast step. Although we did not provide analysis on accuracy of estimating the copula, the extensive experiment results show the effectiveness of our method.
> > >
> > > [2] Stankeviciute, Kamile, Ahmed M Alaa, and Mihaela van der Schaar. "Conformal time-series forecasting." Advances in Neural Information Processing Systems 34 (2021): 6216-6228
> > >
> > > > Instead, use the Guassian as a score function for ICP as ICP works with any score function
> > >
> > > > Figure 11(b) demonstrates that if the intervals are correctly chosen based on ICP, “Direct Gaussian” may provide a smaller interval size than the proposed approach, while satisfying the coverage guarantee simply due to ICP.
> > >
> > > We have implemented the direct Gaussian + CP baseline that uses log likelihood as nonconformity score, please see the results in the next comment.
> > >
> > > Because the RNN underestimates variance as highlighted in previous response, the nonconformity scores ended up being very high. In our experiments, the 90% quantile for calibration set log likelihoods are on the magnitude of -100, that to cover enough samples the regions need to be 10 or 12 times the variance for each variable, resulting in very large uncertainty areas.
> > >
> > > CP is often criticized for producing large uncertainty areas due to its generality. Making distribution assumptions will lead to sharper probabilistic forecasts. Bayesian methods are found to sacrifice calibration (by overfitting to samples) for better accuracy [3]. To combine them (at least naively as we have done) compromises the generality of CP, and loses the sharpness advantages of Bayesian methods. This motivates re-calibration works in Bayesian forecasting, and more sophisticated CP algorithms such as ours.
> > >
> > > [3] Guo, Chuan, Geoff Pleiss, Yu Sun, and Kilian Q. Weinberger. "On calibration of modern neural networks." In International conference on machine learning. PMLR, 2017
> > >
> > > We hope our explanation is clear this time. Many thanks again to the reviewer for engaging in conversation with us.

---

> > > ### Author Response · Authors · 2022-11-21
> > > **Data for additional baseline**
> > >
> > >
> > > Particle dataset $\sigma = 0.01$, with confidence $1-\alpha = 0.9$
> > > | Method | Coverage (90%) | Area $\downarrow$ |
> > > | --- | ----------- | -- |
> > > | L2-Conformal | $88.5 \pm 0.4$ | $7.21 \pm 0.35$ |
> > > | Direct Gaussian| $11.9 \pm 0.09$ | $0.07 \pm 0.31$ |
> > > | Direct Gaussian + CP | $90.1 \pm 0.2$ | $57.7 \pm 4.9$ |
> > > | CF-RNN |  $97.0 \pm 2.3$ | $3.13 \pm 3.24$ |
> > > | Copula-RNN | $\textbf{91.5} \pm 2.1$ | $\textbf{1.06} \pm 0.36$ |
> > >
> > > Particle dataset $\sigma = 0.05$, with confidence $1-\alpha = 0.9$
> > > | Method | Coverage  (90%) | Area  $\downarrow$|
> > > | --- | ----------- | -- |
> > > | L2 |  $89.7 \pm 0.6$  | $7.21 \pm 0.35$ |
> > > | Direct Gaussian| $0.0 \pm 0.0$ | $0.08 \pm 0.02$ |
> > > | Direct Gaussian + CP | $89.8 \pm 0.2$ | $58.6 \pm 4.3$ |
> > > | CF-RNN |  $97.0 \pm 2.3$ |  $5.79 \pm 0.51$ |
> > > | Copula-RNN | $\textbf{90.3} \pm 0.7$ | $\textbf{4.50} \pm 0.07$ |

---

> ### Author Response · Authors · 2022-11-18
> **Response to reviewer FqUe (part 2)**
>
>
> ### Response to clarity
>
> > Page 4: The citation (Gibbs & Candes, 2021) in the first paragraph is not appropriate; the cited paper attacks online learning, but the submitted paper does not consider the same setup, making readers confusing.
>
> Gibbs & Candes also consider conformal prediction for time series forecasting, which is related. However, their setting treats each time step independent (as in online learning) whereas we consider the correlations among multi-step jointly.
>
>
> In the conformal prediction section of related works, we have elaborated on this distinction.
>
>
> > Page 5: why is (8) true? The Sklar’s theorem only implies the first equality.
>
>
> The first equality, by Theorem 1 Sklar's theorem, we have
> $$
> F(s_1, \ldots, s_k) = C(F_1(s_1), \ldots, F_k(s_k))
> $$
>
> The $F_h$ are CDF functions and $F_h(s_h)$ is the probability of on time step $h$, the nonconformity score being less than $s_h$. The $\alpha_h$ values here are introduced as variables.
> $$F_h(s_h) = 1-\alpha_h, \; h \in 1, \ldots, k$$
>
> The last part of the equation is the objective for conformal prediction. We want to find the $\alpha_1, \ldots, \alpha_k$ values such that
>
> $$
>  C(1-\alpha_1, \ldots, 1-\alpha_k)  \geq 1 - \alpha
> $$
>
> In practice, we perform the search in the case of equality, hence we have written (8) as it is. Your comment made us realize that it might be confusing, so we have updated section 4.2 of the paper to clarify.
>
> >Page 5: epsilon_t below the (8) is not defined.
>
> Thank you for pointing this out. corrected to $\alpha_h$.
>
> >Page 7: in “Metrics”, the expectation in the definition of coverage does not make sense to me.
>
> As written in our metric section, coverage is calculated as
> $$coverage_{1-\alpha} =  \mathbb{E}_{x,y \sim \mathbf{X} \times \mathbf{Y}} P(\mathbf{y} \in \Gamma^{1-\alpha}(\mathbf{x})) \approx \frac{1}{n} \sum^n_1 \mathbb{1} (\mathbf{y}_n \in \Gamma^{1-\alpha}_n(\mathbf{x}_n))$$
>
> Which is the expectation over $\mathbf{X} \times \mathbf{Y}$ forthe probability of our prediction region covering $\mathbf{y}$.
>
> > For the Covid19 dataset, it consists of 380 sequences; how can it be collected in UK regions? It would be better if the paper is self-contained. Is there any chance that the exchangeability assumption can be violated for this dataset?
>
> Please see appendix C.2 for a detailed description of how the dataset is obtained and processed. The 380 sequences are daily new cases from 380 regions (to name a few: Barnet, Worthing, East Devon.)  We follow the experiment setup in  Stankeviciute et al. (2021) [3] treat them as interrelated time series that are exchangeable.
>
> [3] Stankeviciute, Kamile, Ahmed M Alaa, and Mihaela van der Schaar. "Conformal time-series forecasting." Advances in Neural Information Processing Systems 34 (2021): 6216-6228
>
>
> Thank you again for reviewing our paper and engaging in discussions with us! Please let us know if anything is still unclear.

---

### Official Review · Reviewer_oH8D · 2022-10-23

**Confidence:** 4
**Correctness:** 3
**Technical Novelty And Significance:** 3
**Empirical Novelty And Significance:** 3
**Recommendation:** 6

**Clarity, Quality, Novelty And Reproducibility:**

Except for the aspects mentioned above in strength and weaknesses, the paper is overall clear for people familiar with the field and the problem. It seems also accessible to a wider audience, although I may be biased on this aspect, being familiar with the topic.

As far as I can tell, the work is original and builds upon previous techniques and papers to present something new with convincing experiments.

**Details Of Ethics Concerns:**

No concerns

**Strength And Weaknesses:**

+: a new method to solve a difficult problem, easily applicable, which appears sound from what I understood.

+: experimental results seem to be good, and are accompanied with code (that I could not run on my laptop in due time, but in reason of my install and the lateness of my review... I will update if I manage to run it in the next days)

-: many typos in the paper, suggesting a rushed in submission. Among others:
* exchangability should be exchangeability (numerous times)
* serires --> series
* equation 1 is an $L_1$ norm, not a $L_2$. Also, if I am correct non-conformity scores in split conformal should be computed on $\mathcal{D}_{cal}$
* equation 4, $A(x_{n+1},\hat{f})$ should be $A((x_{n+1},y),\hat{f})$ to be consistent with earlier notations
* time-step is written time-step, timestep, time step at different places
* joined CDF --> joint CDF
* P5, there is an $\epsilon_t$, which should probably be $\alpha_t$ or $1-\alpha_t$
* Citations are sometimes inconsistent. Format Name (year) should be when the name is supposed to be read, while (name, year) should be used when the name should not be read. P5, Ruschendorf should not be read, hence (name, year) (\citep) should be used. This happens in other places.
* P7: forgotten parenthesis in the definition of coverage$_{1-\alpha}$

-: in the case of auto-regressive, it is unclear how is treated the newly produced point in order to produce further predictions. In particular, since the newly predicted point is a conformal sets, do authors run conformal prediction using weak-labels/intervals (as in "Cauchois, M., Gupta, S., Ali, A., & Duchi, J. (2022). Predictive Inference with Weak Supervision."), or do they simply use the new point-wise prediction? What guarantees that the previous validity guarantees with respect to the ground-truth will hold (as the prediction is not the true value)? Also, how does one obtain the cumulative distribution for the non-conformity scores of the k+1 prediction? Does one use the same cumulative distribution, and if this is the case, why do we need to re-run an optimisation?

-: While the optimisation program introduced to find suitable time-step wise confidence degrees seems fine, it is unclear what we can/should expect from it? In particular, shall we in practice have very similar confidence degrees or very imbalanced ones? What justify to have different confidence degrees at each run? Why not optimizing to minimize the obtained volume? How does it compare to the constant time dichotomic search assuming all confidence degrees being equal?

**Summary Of The Paper:**

The paper describes a new conformal method to perform guaranteed prediction in multi-step multi-variate time series. In order to achieve better calibration and efficiency across dimensions, it proposes the use of copulas.

**Summary Of The Review:**

The paper introduces a new method to perform multi-dimensional, multi-step ahead time series prediction. Experiments performed on various data sets are convincing, but remain limited in the number of dimensions predicted (at most 3 dimensions in the case of drone trajectory prediction). This is however fine, as the focus is on multi-step time series prediction.

Authors may be interested in knowing that the approach of Johnstone and Cox has been recently adapted to cope with normalized scores, hence providing adaptive ellipsoids (see "Messoudi, S., Destercke, S., & Rousseau, S. Ellipsoidal conformal inference for Multi-Target Regression.").

---

> ### Author Response · Authors · 2022-11-18
> **Response to Reviewer oH8D**
>
> We sincerely thank reviewer oH8D for providing thorough and insightful comments.
>
> > -: many typos in the paper, suggesting a rushed in submission. Among others:
>
> We apologize for the typos! The updated version has them all corrected. Thank you for carefully reading our paper.
>
> > -: in the case of auto-regressive, it is unclear how is treated the newly produced point in order to produce further predictions. In particular, since the newly predicted point is a conformal sets, do authors run conformal prediction using weak-labels/intervals (as in "Cauchois, M., Gupta, S., Ali, A., & Duchi, J. (2022). Predictive Inference with Weak Supervision."), or do they simply use the new point-wise prediction? What guarantees that the previous validity guarantees with respect to the ground-truth will hold (as the prediction is not the true value)? Also, how does one obtain the cumulative distribution for the non-conformity scores of the k+1 prediction? Does one use the same cumulative distribution, and if this is the case, why do we need to re-run an optimisation?
>
> In our current setup we only use the point-wise predictions for autoregressive forecasting. The coverage guarantee will always only be on the $k$ step that are predicted by the model, whether it is time steps $\{t+1, \ldots, t+k\}$ or $\{t+2, \ldots, t+k+1 \}$, etc. The algorithm described in the paper does not give cumulative coverage guarantees for over $k$ steps of forecasts.
>
> We assume we have validation data for the length of the autoregressive forecasts. This way, we model the copula for uncertainty of _the first $k$ steps of the autoregressive forecast_, _the $t+1$ to $t+k+1$ steps of the autoregressive forecast_, and so on. As outline in paragraph 2 of section 4.3 in the paper, we might or might not need to refit to every one of these cases.
>
> We understand that it's desirable to have coverage guarantees for the entire forecasting horizon, or use predicted intervals for future predictions. However these goes beyond the scope of our paper, so we leave it for future work.
>
>
> > -: While the optimisation program introduced to find suitable time-step wise confidence degrees seems fine, it is unclear what we can/should expect from it? In particular, shall we in practice have very similar confidence degrees or very imbalanced ones? What justify to have different confidence degrees at each run? Why not optimizing to minimize the obtained volume? How does it compare to the constant time dichotomic search assuming all confidence degrees being equal?
>
> Please see the newly added appendix C.6 for a study on finding $\alpha_h$.
>
> Thank you again for the thoughtful questions and comments! Please let us know if any of the explanations are unclear.

---

### Official Review · Reviewer_aA8r · 2022-10-26

**Confidence:** 2
**Correctness:** 4
**Technical Novelty And Significance:** 3
**Empirical Novelty And Significance:** 3
**Recommendation:** 6

**Clarity, Quality, Novelty And Reproducibility:**

The derivation is readable and reasonable. The proposed uncertainty quantification algorithm is novel and achieves the expected performance. Besides, the given algorithm which summarizes the CoupulaCPTS procedure helps reproduce the method effectively.


**Strength And Weaknesses:**

Pros:

(1) The proposed method is novel and the proof is solid.

(2) The conducted experiments are comprehensive and show outstanding performance.

(3) The concerned problem has practical applications in the prediction model.

Cons:

(1) Some case studies can be added to interpret its detailed application better.


**Summary Of The Paper:**

Summary of the paper
In this paper, the authors present a calibrated and efficient conformal prediction algorithm for multi-step time series forecasting and utilize copulas to model the dependency between forecasted time steps. The results show the proposed method performed well on synthetic and real-world datasets both in calibration and efficiency.

Contributions

(1) The proposed method is a general uncertainty quantification algorithm that has wide applications.

(2) CopulaCPTS shows competitive performance over other baselines and the experiment is sufficient.
Strength and weaknesses

**Summary Of The Review:**

In this paper, the authors present a method to quantify the uncertainty of the prediction model, which is novel and meaningful. The designed experiments are efficient and achieve the expected results. Hence, this is a good paper that could be accepted.

---

> ### Author Response · Authors · 2022-11-15
> **Clarification request on your review**
>
> Thank you for recognizing the value of our work, and the helpful suggestion.
>
> To clarify, can you provide more details on what type of case studies you are asking for? We can add accordingly.

---

### Official Review · Reviewer_gQHv · 2022-10-28

**Confidence:** 2
**Correctness:** 4
**Technical Novelty And Significance:** 2
**Empirical Novelty And Significance:** 2
**Recommendation:** 6

**Clarity, Quality, Novelty And Reproducibility:**

The methodology and thoery is generally clear, but the settings are not well covered from the time series forecasting literatures.

- in eq (7) , h is typo?
- the data split of training and val is not clearly described, especially multivatie settings.

**Strength And Weaknesses:**

## Strength
- develop more efficient conformal prediction method with the simple yet effective idea of Coupla.
- clear motivation and idea
- extensive experiments are done



## Weakness
- multivariate parts that author claim to be new seems to not be really in the novel parts, but multi-step ones are main focus. Otherwise, kindly describe why existing methods by Stankevicˇiu ̄te ̇ et al. (2021) cannot handle this setting (with small modifications).
- enhance UQ literatures and discussion on other classes. For example there are several advances in forecasting based on quantile regresison methods for univariate and multivariate https://arxiv.org/abs/2202.11316,https://arxiv.org/abs/2111.06581, and even conformal prediction enhance quantile regression method itself https://arxiv.org/abs/1905.03222.
- more deep dive on behvarious ${\alpha_t\}$. For example, is this related to the author claim the capability of capturing dependency
- the experiment setting are poorly described and difficult to examine univatie/multivate cases and which various base forecasters are used and why.


**Summary Of The Paper:**

This paper proposes conformal prediction method for multivariate multistep TS forecasting.

**Summary Of The Review:**

This work proposed simple and yet effective method for multi-step forecasitng,  even with several unclear descriptions like general forecasting settings and experiment settings.

---

> ### Author Response · Authors · 2022-11-18
> **Response to Reviewer gQHv**
>
>
> We would like to sincerely thank the Reviewer gQHv for providing valuable comments and recognizing the value of our work.
>
> ### Response to Weaknesses
>
> > W1: multivariate parts that author claim to be new seems to not be really in the novel parts, but multi-step ones are main focus. Otherwise, kindly describe why existing methods by Stankeviciute et al. (2021) cannot handle this setting (with small modifications).
>
> We agree with review gQHv's comment. In Stankeviciuteet al. (2021), only univariate time series was handled by the algorithm; the multivariate setting was mentioned as a future direction. We made a small modification to cover multivariate data by computing the vector norm in Eqn (1).
>
> Our key contribution is multi-step forecasting. Due to the curse of dimensionality, the uncertainty regions often becomes very large for multivariate time series, if we directly apply CP or Stankeviciuteet al. (2021)'s approach. We developed our method CopulaCPTS to significantly shrink the uncertainty regions while still maintaining coverage, especially for multivariate datasets (see table 1). To summarize, we mention multivariate as a motivating setting and useful application area for this work.
>
> > W2: enhance UQ literatures and discussion on other classes. For example there are several advances in forecasting based on quantile regresison methods for univariate and multivariate https://arxiv.org/abs/2202.11316,https://arxiv.org/abs/2111.06581, and even conformal prediction enhance quantile regression method itself https://arxiv.org/abs/1905.03222.
>
> Thank you for the valuable suggestion! We have incorporated these papers in our literature review section.
> Even though quantile methods are widely used in time-series forecasting UQ, they do not give finite sample guarantees on coverage, so they are not directly related to this work. In the CP section we have discussed conformal quantile prediction (Romano et al., 2019) and cited a later work (Sousa et al. 2022).
> Please understand that in the interest of space, we have kept the discussion concise.
>
>
> > W3: more deep dive on behvarious $\alpha$. For example, is this related to the author claim the capability of capturing dependency?
>
> Please see the updated Appendix C.6 for a study on $\alpha_h$. Let us know if it clarifies the role $\alpha_h$ plays in representing the correlation.
>
> > W4: the experiment setting are poorly described and difficult to examine univatie/multivate cases and which various base forecasters are used and why.
>
> We have edited section 5.1, section 5.2, and Appendix C.1 for better clarity on experimental setting and model selection.
>
>
> ### Response to Clarity
> > in eq (7) , h is typo?
>
> Yes, thank you for the note. We've corrected it in the updated version of the paper.
>
> > the data split of training and val is not clearly described, especially multivatie settings.
>
> See W4 for the updated description.
>
>
> Thank you again for your thorough review and insightful questions! Let us know if anything remains unclear.

---

### Decision · Program_Chairs · 2023-01-20

**Decision:**

Reject

**Justification For Why Not Higher Score:**

The overall idea is sound and practical. But the proof of validity has an issue that I mentioned in the meta-review. That is why the paper is recommended not to be accepted in its current form.

**Justification For Why Not Lower Score:**

N/A

**Metareview: Summary, Strengths And Weaknesses:**

This paper addresses a method for multi-step time series forecasting where the copula is employed to model the dependency between forecasted times steps, improving the efficiency in conformal prediction. Recently the conformal prediction draws attention and its extensions to various cases have been developed. The conformal prediction can be directly applied to the single-step time series forecasting without considering the temporal dependency. In multi-step prediction, modeling between time steps is crucial and the copula is one easy and practical way to link marginals with the joint distribution. All reviewers agree that the idea is timely and sound. However, there is one critical concern on the rigorousness, which is not resolved even after the author-reviewer discussion. The marginal coverage probability is taken with respect to the test data as well as the copula which is calculated using the calibration set. The proof of validity should consider the randomness from the calibration set, which is not clearly described or not clearly responded by the authors. One possible way to overcome this is to use an additional hold-out set to estimate the copula, which does not need to introduce any correction (including Bonferroni correction). This requires more samples to run this idea, compared to other approaches. The experiments well support the efficiency of the method but adding theoretical justification or at least insights in some ways will be merits.
Therefore, the paper is not recommended for acceptance in its current form. I hope authors found the review comments informative and can improve their paper by addressing these carefully in future submissions.


**Summary Of Ac-Reviewer Meeting:**

I did not have a virtual meeting with all reviewers. However, there were extensive discussions with them to pull out the decision. Moreover, I was able to get a useful and direct feedback from particular reviewers whose comments are critical.